# SATCH: Specialized Assistant Teacher Distillation to Reduce Catastrophic Forgetting

## Abstract

Continual learning enables models to learn new tasks sequentially without forgetting previously learned knowledge. Knowledge distillation reduces forgetting by using a single "teacher" model to transfer previous knowledge to the "student" model. However, existing methods face challenges, specifically loss of task-specific knowledge, limited diversity in the transferred knowledge, and delays in teacher availability. These issues stem from self-distillation, where the teacher is a mere snapshot of the student after learning a new task, inheriting the student's biases and becoming available only after learning a task. We propose Specialized Assistant TeaCHer distillation (SATCH), a novel method that uses a smaller assistant teacher trained exclusively on the current task. By incorporating the assistant teacher early in the learning process, SATCH provides task-specific guidance, improves the diversity of transferred knowledge, and preserves critical task-specific knowledge. Our method integrates seamlessly with existing knowledge distillation techniques, and experiments on three standard continual learning benchmarks show that SATCH improves accuracy by up to 12% when combined with four state-of-the-art methods. Code is available in supplementary materials.

## 1 Introduction

Continual learning aims to incrementally learn a sequence of tasks (Chen & Liu, 2018). Typically, once a task is learned, its training data is no longer accessible. However, when learning a new task, models are prone to catastrophic forgetting, where they experience a significant drop in accuracy on previously learned tasks. Forgetting occurs because the parameters associated with prior tasks are overwritten when learning a new task (McCloskey & Cohen, 1989). Catastrophic forgetting is particularly challenging in class incremental learning, where the model must learn a sequence of tasks without access to task identifiers (Van de Ven et al., 2022).

Knowledge distillation is commonly used in continual learning to minimize catastrophic forgetting. These methods introduce a regularization term that encourages the "student" model to mimic the outputs of its previous version, known as the "teacher". Typically, the teacher is created by cloning the student before it learns a new task, which results in both models sharing the same backbone and initialization (Li & Hoiem, 2017). To further reduce forgetting, these methods often integrate replay buffer selection (Rebuffi et al., 2017; Ahn et al., 2021), where samples from previous tasks are stored in a buffer and replayed during training to improve the retention of old task knowledge.

While knowledge distillation methods have shown promise in mitigating catastrophic forgetting, they face three key challenges: **(1) Limited diversity in knowledge transferred**: Knowledge distillation methods often rely on a single main teacher model with the same architecture and initialization as the student, limiting the knowledge transfer diversity. Although multi-teacher approaches using different random seeds (Li et al., 2023) or diverse datasets (Wu et al., 2019) have been explored, they often require full data access, which is impractical in real-world settings with limited data availability. **(2) Forgetting task-specific knowledge**: As the student model learns new tasks, there is a shift in weights, causing it to forget previously learned task-specific information, especially in class incremental learning where task identifiers are unavailable during inference (Van de Ven et al., 2022). Forgetting is compounded because the main teacher, being a snapshot of the student after learning a task, inherits the student's forgotten knowledge. Replay buffer methods can help mitigate forgetting by storing samples of previous tasks, but they are constrained by the limited amount of data stored.

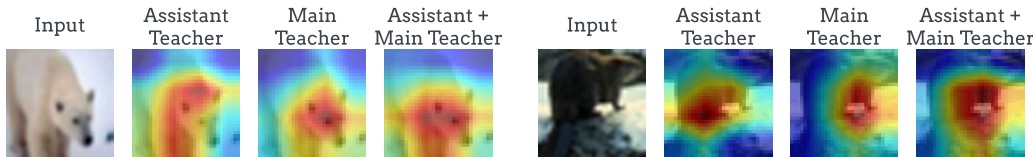

Figure 1: Grad-CAM (Chattopadhay et al., 2018) visualization of the different features for SATCH assistant teacher, main teacher (DER++), and combined assistant + main teacher for CIFAR100.

**(3) Limited use of main teacher knowledge**: Main teacher knowledge is typically only available after the student has learned the new task, restricting the benefits of using soft labels to guide student learning (Tzeng et al., 2015) and filtering samples stored in the replay buffer which is important in real-world noisy scenarios (Sarfraz et al., 2023). While training an additional teacher prior to the student learning the new task has been explored (Hou et al., 2018; Kim et al., 2023), these methods primarily focus on guiding new task learning. These challenges raise two important questions: first, *how can we diversify and preserve task-specific knowledge transferred from the main teacher?* and second, *how can we use this knowledge to improve student accuracy in real-world noisy scenarios?*

To address these challenges, we propose Specialized Assistant TeaCHer distillation (SATCH), which uses a smaller assistant teacher specialized in learning task-specific knowledge to improve knowledge distillation. SATCH introduces two key innovations: First, to capture task-specific complementary knowledge, the assistant teacher is designed to use a smaller backbone, focusing exclusively on learning a single task. This enables the assistant teacher to capture task-specific features that differ from those learned by the main teacher, as shown in Figure 1. For example, with the bear input, the assistant teacher focuses on features distinct from those of the main teacher. When their outputs are combined, the resulting feature map is broader. Second, the assistant teacher is trained on the new task before the student learns the new task (Hou et al., 2018), allowing the assistant teacher to guide the student. Throughout the learning process, the assistant teacher helps select buffer samples, which is important in noisy scenarios (Sarfraz et al., 2023), by filtering out noisy labels and identifying representative samples through comparison with the student's predictions.

In SATCH, the assistant teacher's knowledge is used in three ways: **(1) Guide new task learning**: The assistant teacher generates soft labels to guide learning, ensuring the student learns inter-task relationships when learning new tasks. **(2) Refine buffer selection**: During buffer selection, the assistant teacher's prediction is compared with the student's prediction to identify samples where both models agree, retaining more representative samples and filtering out noise. **(3) Diversify distilled knowledge**: The assistant teacher's specialized knowledge is stored alongside the buffer samples to provide a complementary view of previous knowledge. The specialized knowledge is combined with the main teacher's generalized knowledge by averaging the relevant output logits. These three components enable SATCH to use the complementary task-specific knowledge from the assistant teacher to reduce catastrophic forgetting.

Our contributions are as follows: (1) We propose SATCH, a novel continual learning method that incorporates a specialized assistant teacher into existing knowledge distillation methods to provide task-specific guidance, provide complementary knowledge, and preserve task-specific knowledge. (2) We introduce three key components that use the complementary task-specific knowledge: guiding new task learning, refining buffer selection, and diversifying distilled knowledge. (3) We validate the effectiveness of SATCH on three benchmark datasets and four state-of-the-art knowledge distillation methods, demonstrating its robustness in class incremental learning and noisy environments.

## 2 RELATED WORK

Continual learning methods span several directions, notably regularization, rehearsal, parameter isolation, and dynamic architecture methods (De Lange et al., 2021).

**Knowledge Distillation Methods**: Knowledge distillation was originally developed to transfer knowledge from a large teacher network to a smaller student network (Hinton, 2015; Zagoruyko & Komodakis, 2017). In continual learning, knowledge distillation prevents forgetting by encourag-

ing the current model to mimic the outputs of a previous version of itself. This is commonly applied through self-distillation (Mobahi et al., 2020), where the main teacher and student share the same backbone, with the main teacher being a copy of the student before learning a new task. To further reduce forgetting, these methods often include a replay buffer that stores past samples and replays a subset when learning a new task. DER++ (Buzzega et al., 2020) introduces an additional distillation loss by storing the model outputs of old knowledge with the samples and enforcing output space consistency during learning. SSIL (Ahn et al., 2021) separates the softmax layer and performs task-specific distillation using a previous copy of the student. Other methods, such as ESMER (Sarfraz et al., 2023) and CLS-ER (Arani et al., 2022), focus the main teacher model on retaining old knowledge and enforcing output consistency during learning. However, these methods often struggle to retain task-specific knowledge since the main teacher inherits the forgotten previous knowledge, and there is only a single teacher, which limits the diversity of distilled knowledge.

**Multi-teacher Distillation**: Multi-teacher distillation aims to improve student generalization by inheriting diverse knowledge from multiple teachers (Gou et al., 2021; Wang & Yoon, 2021). Teachers are generated using different random seeds (Li et al., 2023), trained on different datasets (Wu et al., 2019), or designed with varying backbones (You et al., 2017). However, multi-teacher methods often require full data access, which may not be feasible in real-world applications with limited data availability (Chaudhry et al., 2019).

**Auxiliary Network Methods**: While knowledge distillation methods focus on preventing forgetting, some methods also aim to improve new task learning. Hou et al. (2018) train an auxiliary teacher before the student begins learning a new task, distilling soft labels to the student as it learns a new task. Similarly, Kim et al. (2023) incorporates new task knowledge by training a copy of the student on the new task, using it as an additional regularization term during the student's learning. Although these methods incorporate new task knowledge, they create the auxiliary teacher by simply copying the student, which makes the model prone to losing task-specific knowledge.

**Dynamic Architecture and Pre-trained Methods**: Dynamic architecture methods (Yan et al., 2021; Lin et al., 2024) add additional parameters for each task and use a task-ID predictor to dynamically select the task relevant parameters during inference. Similarly, pre-trained methods (Wu et al., 2022; Huang et al., 2025) use pre-trained models to extract generalized features with additional parameters added to capture task-specific features. These approaches achieve state-of-the-art accuracy. However, they face challenges with restricted memory due to the linear growth of parameters with the number of tasks. Furthermore, they can be less effective in real-world settings where there is overlapping classes (Mi et al., 2020) or noisy labels (Sarfraz et al., 2023).

Our proposed method, SATCH differs from these auxiliary approaches by using an assistant teacher with a smaller backbone who specializes in a single task. SATCH's design improves task-specific knowledge retention by modifying the main teacher's output logits, allowing seamless integration with existing knowledge distillation methods. Additionally, the assistant teacher's complementary knowledge helps filter out noisy samples, a feature not addressed by other auxiliary methods, making SATCH more robust to real-world noisy scenarios.

## 3  SPECIALIZED ASSISTANT TEACHER DISTILLATION

Our approach improves the diversity of knowledge transferred by introducing a smaller, task-specific assistant teacher, which plays a critical role in three key components: guide new task learning, refine the buffer selection process, and diversify distilled knowledge, as seen in Figure 2.

The class incremental learning classification problem is divided into $T$ tasks. For each task $t \in \{1, \ldots, T\}$, input samples $x$ and their corresponding ground truth labels $y$ are drawn from the task-specific distribution $\mathcal{D}_t$. The objective is to learn a model that approximates the joint distribution over all tasks, enabling it to classify the observed classes at inference without the task identity. At each training step, the model receives a batch of labeled samples $(x_t, y_t) \sim \mathcal{D}_t$ from the current task, along with memory samples $(x_t^m, y_t^m) \sim \mathcal{M}$ drawn from an episodic memory buffer. The student model, $\theta_{1:t}^s$, produces output logits $z_{1:t}^s$ for tasks 1 to $t$, given an input $x \in \mathcal{D}_t \cup \mathcal{M}$. To focus on the output logits corresponding to task $t$, a task-specific binary mask $m_t$ is applied to the output logits $z_{1:t}^s$ via an element-wise operation $\odot$, resulting in $z_t^s = m_t \odot z_{1:t}^s$. The binary mask $m_t$ masks out the logits that do not correspond to task $t$.

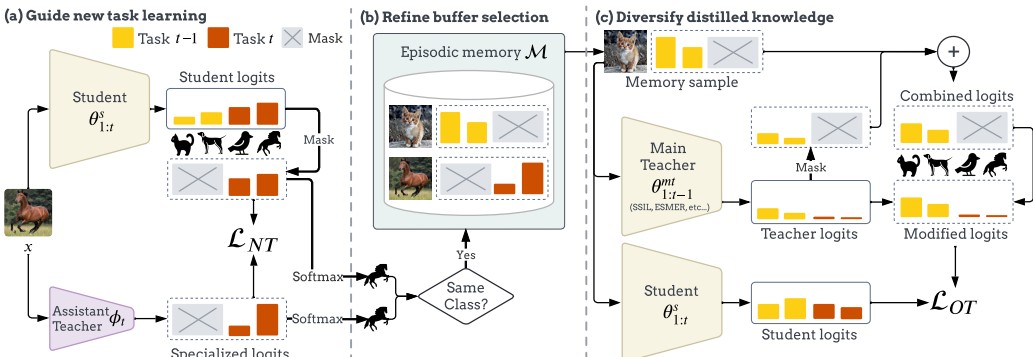

Figure 2: Overview of SATCH components after training the assistant teacher $\phi_t$ for task $t$: (a) When learning a new task $t$, an input image $x$ is classified by the student model $\theta^s_{1:t}$, which produces task-specific logits after applying mask $m_t$. The assistant teacher $\phi_t$ generates soft labels for $x$ to guide learning through new task loss $\mathcal{L}_{NT}$. (b) After generating predictions for input image $x$, the predicted labels of the student and assistant teacher are compared to refine the buffer samples, with the assistant teacher's logits also stored in the memory buffer $\mathcal{M}$. (c) To reduce forgetting of past tasks, past task samples and their assistant teacher logits are replayed from memory. The sample is classified by the main teacher model $\theta^{mt}_{1:t-1}$, which is then masked and combined with the assistant teacher logits. The modified logits are used for distillation in the old tasks loss $\mathcal{L}_{OT}$.

### 3.1 TRAINING THE ASSISTANT TEACHER

The assistant teacher improves knowledge diversity through its smaller backbone network and specialization in a single task. For instance, if the main teacher uses a ResNet-18 backbone, the assistant teacher uses a reduced ResNet-18 (Lopez-Paz & Ranzato, 2017), with fewer feature maps across all layers. While the generalized main teacher balances learning across multiple tasks, the assistant teacher focuses on capturing task-specific feature representations. For each new task $t$, the assistant teacher $\phi_t$ is trained using cross-entropy loss (Rebuffi et al., 2017), which is then used to guide the student model $\theta^s_t$ during task learning and assist in filtering noisy samples during buffer selection.

We extend the episodic memory buffer $\mathcal{M}$ to manage memory efficiently. When learning a new task, samples from the current task, $x_t$, and their labels, $y_t$, are stored in the replay buffer $\mathcal{M}$. In SATCH, we extend the buffer to store the output logits generated by the assistant teacher. The samples stored in memory contain tuples of the form $(x^m_t, y^m_t, z^{\phi_t}_t)$, where $x^m_t$ is a memory sample for task $t$, $y^m_t$ is its label, and $z^{\phi_t}_t$ is the output logits from the assistant teacher $\phi_t$ given the stored input $x^m_t$. This approach allows us to maintain a single assistant teacher throughout the learning process.

### 3.2 GUIDE NEW TASK LEARNING

Traditional knowledge distillation methods typically rely on one-hot encoded labels when learning. However, soft labels capture inter-class relationships and guide the student model towards a more optimal solution space, improving accuracy on new tasks (Tzeng et al., 2015). Our component, guide new task learning (NEWL), is formalized through the following loss function:

$$\mathcal{L}_{NT} = \mathcal{L}_{CE}(z^s_{1:t}, y) + \lambda \mathcal{D}_{KL}(p^\tau(z^s_{1:t}) \parallel p^\tau(z^{\phi_t}_{1:t})), \tag{1}$$

where $\mathcal{L}_{NT}$ is the total loss for the new task, $\mathcal{L}_{CE}$ denotes the cross-entropy loss between the student model's logits $z^s_{1:t}$ and the ground truth $y$, and $\mathcal{D}_{KL}(\cdot \parallel \cdot)$ is the Kullback-Leibler divergence. Here, $p^\tau(\cdot)$ represents temperature-scaled probability vectors for the student logits $z^s_{1:t}$ and the assistant teacher logits $z^{\phi_t}_{1:t}$, with $\tau$ as the temperature scaling parameter. The hyperparameter $\lambda$ controls the guidance from the assistant teacher, transferring task-specific knowledge to the student.

Since the assistant teacher $\phi_t$ is trained exclusively on task $t$, distilling logits from tasks 1 to $t$ can result in forgetting when $t > 1$. To address incorrectly distilling irrelevant knowledge, we use task-wise distillation (Zhao et al., 2022) to distill only the relevant logits from task $t$, removing the risk of incorrect distillation of previous tasks. Specifically, we apply a mask $m_t$ to keep only task-specific

logits from task $t$ for the main and assistant teachers during distillation. Additionally, we implement task-recency bias mitigation (BIAM) by using only the current task $t$ logits in the cross-entropy loss, preventing task-recency bias that can cause forgetting of older knowledge due to the imbalance of newly learned samples (Ahn et al., 2021). The resulting loss function using the task-specific logits:

$$\mathcal{L}_{NT} = \mathcal{L}_{CE}(z_t^s, y) + \lambda \mathcal{D}_{KL}(p^\tau(z_t^s) \parallel p^\tau(z_t^{\phi_t}))), \tag{2}$$

where $z_t^s$ refers to the student's logits for task $t$, and $z_t^{\phi_t}$ are the logits of the assistant teacher for task $t$. Using the assistant teacher's soft labels helps guide the student toward an optimal solution space, improving accuracy on the new task. Additionally, focusing only on the relevant logits for task $t$ reduces task-recency bias, which reduces forgetting.

## 3.3 REFINE BUFFER SELECTION

In real-world scenarios, noisy labels are common in continual learning (Kim et al., 2021), and knowledge distillation methods suffer from accuracy loss due to the storage and replay of noisy samples (Sarfraz et al., 2023). To reduce the impact of noisy labels, we use a buffer selection strategy that uses the specialized assistant teacher $\phi_t$ to provide an alternative understanding for identifying representative samples. Our Refine Buffer Selection (BUFS) component refines Reservoir Sampling (Vitter, 1985) by incorporating the knowledge of both the assistant teacher $\phi_t$ and the student $\theta_{1:t}^s$ during the pre-selection process. The specialized assistant teacher and generalized student are trained on different backbones, allowing them to interpret samples from different perspectives.

To isolate the classification to task-specific knowledge, we apply a mask $m_t$ to the logits of the student and the assistant teacher before applying softmax to compute the predicted labels. To determine if a sample is stored or discarded, we use a student and teacher agreement criterion:

$$(x_i^m, y_i^m, z_i^{\phi_t}) = \{(x_i, y_i, z_i^{\phi_t}) \mid (x_i, y_i) \in D_t, \hat{y}_i^s = \hat{y}_i^{\phi_t}\}, \tag{3}$$

where $(x_i^m, y_i^m, z_i^{\phi_t})$ are memory samples selected for task $t$. A sample $(x_i, y_i)$ is stored in the memory buffer $\mathcal{M}$ if the student and assistant teacher predict the same label. The logits $z_{1:t}^s$ and $z_{1:t}^{\phi_t}$ are masked with $m_t$ to focus on task-relevant outputs, and softmax is applied to compute the predicted labels $\hat{y}_i^s$ and $\hat{y}_i^{\phi_t}$ for the student and assistant teacher, respectively. By using the different views of the assistant teacher and the student, we store the most reliable samples, reducing the influence of noisy data and maintaining stable, accurate representations in memory.

## 3.4 DIVERSIFYING KNOWLEDGE DISTILLATION

Inspired by multi-teacher distillation, which has been shown to improve student generalization (Gou et al., 2021; Wang & Yoon, 2021), we propose our component, diversify distilled knowledge (DIVK) to diversify the knowledge transferred by combining the specialized output from the assistant teacher $\phi_t$ with the generalized output from the main teacher. Existing methods (Buzzega et al., 2020; Ahn et al., 2021) only use a single main teacher, which only provides a single understanding of previous knowledge. When a sample is retrieved from the replay buffer, the main teacher produces output logits containing prior knowledge, which the student mimics to help prevent forgetting.

During the knowledge distillation process, samples $(x_t^m, y_t^m, z_t^{\phi_t})$ are retrieved from the replay buffer $\mathcal{M}$, where $(x_t^m, y_t^m)$ is the memory samples, and $z_t^{\phi_t}$ represents the assistant teacher's output logits. The main teacher classifies the memory samples to produce output logits representing the old tasks' knowledge. In SATCH, we diversify the knowledge distilled by combining the assistant teacher's stored logits with the main teacher's output logits. While Song & Chai (2018) suggests averaging the logits, simply averaging would include knowledge the assistant teacher has not learned. To avoid distilling irrelevant information, we apply a task-specific masking process to the main teacher's output logits before combining them:

$$\hat{z}_t^{mt} = \frac{1}{2}(z_t^{mt} + z_t^{\phi_t}), \tag{4}$$

where $z_t^{mt}$ represents the main teacher's output logits for task $t$, and $z_t^{\phi_t}$ are the logits produced by the assistant teacher for task $t$. The averaged logits, $\hat{z}_t^{mt}$, are then incorporated into the main teacher's output for distillation to the student:

$$\hat{z}_{1:t}^{mt} = \bar{m}_t \odot z_{1:t}^{mt} + \hat{z}_t^{mt}, \tag{5}$$

where $\hat{z}_{1:t}^{mt}$ refers to the main teacher's output logits for all tasks learned up to task $t$, $\bar{m}_t$ is the complementary task mask that excludes task $t$, and $m_t$ is the mask for task $t$. The masking process ensures that only the logits for task $t$ are updated, while the logits for other tasks remain unchanged. SATCH integrates seamlessly with existing methods that use different main teacher representations, as it only requires updating the output logits. For example, DER++ (Buzzega et al., 2020) stores the student's output logits in the buffer and distills them during training, while ESMER (Sarfraz et al., 2023) and CLS-ER (Arani et al., 2022) focus the teacher model on retaining old knowledge and distilling output logits to the student. The loss function for retaining previous knowledge is designed to:

$$\mathcal{L}_{OT} = \mathcal{L}_{CE}(z_{1:t}^s, y) + \mathcal{D}_{KL}\left(p^\tau(z_{1:t}^s) \parallel p^\tau(\hat{z}_{1:t}^{mt})\right), \tag{6}$$

where $\mathcal{L}_{OT}$ represents the loss for old tasks, combining the cross-entropy loss $\mathcal{L}_{CE}$ is calculated from a memory sample $x_t^m$ to generate the student's logits $z_{1:t}^s$ and compared with the ground truth $y_t^m$, and the Kullback-Leibler divergence $\mathcal{D}_{KL}$ between the student's probability distribution $p^\tau(z_{1:t}^s)$ and the modified logits $p^\tau(\hat{z}_{1:t}^{mt})$ from the main teacher. Finally, the total loss in the knowledge distillation process is $\mathcal{L}_{total} = \mathcal{L}_{NT} + \mathcal{L}_{OT}$.

# 4 EXPERIMENTS

**Setup**: All methods are evaluated in a class incremental learning setting. Once a task is learned, the training data is discarded except for the data stored in the replay buffer. After learning all tasks, each model is evaluated using the test data without access to the task ID during inference. Following Buzzega et al. (2020), all models use a ResNet-18 backbone network and are trained with an SGD optimizer. SATCH's assistant teacher uses a reduced ResNet-18 backbone network (Kang et al., 2022) unless specified otherwise. Following Lin et al. (2022), we measure the average accuracy and average forgetting of all tasks after learning. We set $\lambda = 0.1$ for all buffer sizes and datasets to reduce the method's dependency on extensive hyperparameter tuning (see Appendix A.9).

**Dataset**: We evaluate our method on four benchmark datasets. A task represents an image classification task for a given group of objects. Following existing continual learning approaches, each dataset's training and testing instances are split evenly into tasks and classes. These datasets are CIFAR100 (Krizhevsky & Hinton, 2009), a visual object dataset split into 10 tasks with 10 different classes; MiniImageNet (Vinyals et al., 2016), a variant of ImageNet (Krizhevsky et al., 2012) with 10 tasks, each containing 10 different classes; TinyImageNet (Le & Yang, 2015), another ImageNet variant with 10 tasks, each including 20 different classes. GCIL-CIFAR100 (Mi et al., 2020), a realistic real-world setting applied onto CIFAR100 with 10 tasks where the number of classes, appearing classes, and their sample sizes for each task are sampled from a probabilistic distribution.

**Baselines**: We compare against upper bound (JOINT), lower bound (SGD), rehearsal-based methods (ER, ER-ACE), regularization method (PASS), and integrate SATCH into baselines using various state-of-the-art knowledge distillation techniques DER++, SSIL, CLS-ER, and ESMER.

## 4.1 RESULTS

We evaluate the accuracy impact of SATCH on various knowledge distillation methods. Table 10 demonstrates the accuracy improvements when combining SATCH with DER++, SSIL, CLS-ER, and ESMER, which we denote as "+ SATCH (ours)". SATCH improves the accuracy of these methods across different datasets and buffer sizes. For example, on CIFAR100 with buffer sizes of 1000 and 5000, combining ESMER with SATCH results in accuracy improvements of 6.54% and 3.19%, respectively. Similarly, on TinyImageNet, the accuracy improvements are 7.85% and 2.39%, and on MiniImageNet, with 7.13% and 2.59%. Also, on GCIL-CIFAR100, a more challenging scenario with overlapping classes, there is an accuracy improvement of 2.51% and 2.20%.

To further understand SATCH's impact on individual task accuracy as more tasks are introduced, we analyze the task accuracy progression for specific tasks with and without SATCH in Figure 3. While the initial task accuracy without SATCH is slightly higher, it declines more rapidly as additional tasks are learned. In contrast, SATCH maintains more stable accuracy over time, showing a smaller decline in performance on past tasks as new ones are introduced. This suggests that SATCH reduces task-recency bias, which often leads to higher accuracy for the most recent task at the cost of forgetting older ones.

Table 1: Comparison of CL methods with varying datasets and buffer sizes. We report accuracy and standard deviation over 5 different runs for each result.

| Memory Size | CIFAR100 | | TinyImageNet | | MiniImageNet | | GCIL-CIFAR100 | |
|---|---|---|---|---|---|---|---|---|
| | 1000 | 5000 | 1000 | 5000 | 1000 | 5000 | 1000 | 5000 |
| JOINT (Upper bound) | 70.11±0.21 | | 59.69±0.13 | | 45.40±0.09 | | 57.21±1.42 | |
| SGD (Lower bound) | 9.34±0.05 | | 8.12±0.08 | | 9.28±0.06 | | 10.04±0.21 | |
| PASS (CVPR 21) | 48.34±0.92 | | 41.18±0.88 | | 36.48±1.12 | | - | |
| ER (Neurips 19) | 26.88±0.81 | 42.64±0.86 | 12.24±0.32 | 25.54±0.44 | 22.22±0.53 | 32.52±0.52 | 22.41±0.39 | 30.62±0.26 |
| ER-ACE (ICLR 22) | 43.58±0.46 | 55.77±0.54 | 26.36±0.21 | 37.86±0.41 | 31.24±0.30 | 35.90±0.41 | 29.89±0.41 | 34.12±0.12 |
| DER++ (Neurips 20) | 44.62±0.56 | 56.39±1.06 | 18.92±0.39 | 36.39±0.45 | 25.81±0.66 | 36.13±0.75 | 30.68±0.37 | 41.32±0.42 |
| + SATCH (ours) | **48.38±0.19** | **59.97±0.18** | **30.54±0.27** | **42.33±0.26** | **33.08±0.51** | **39.95±0.26** | **37.67±0.15** | **44.23±0.11** |
| SSIL (ICCV 21) | 40.70±0.40 | 51.54±0.89 | 33.28±0.12 | 39.13±0.47 | 31.66±0.69 | 33.44±0.76 | - | - |
| + SATCH (ours) | **42.95±0.17** | **54.06±0.64** | **34.33±0.15** | **42.51±0.27** | **33.83±0.57** | **37.20±1.90** | - | - |
| CLS-ER (ICLR 22) | 45.47±0.63 | 59.63±1.12 | 22.67±0.42 | 39.43±0.08 | 32.32±1.29 | 37.56±0.93 | 31.46±0.43 | 40.59±0.55 |
| + SATCH (ours) | **52.36±0.30** | **61.39±0.30** | **37.33±0.19** | **46.49±0.04** | **37.38±0.85** | **41.88±0.38** | **36.12±0.21** | **42.95±0.41** |
| ESMER (ICLR 23) | 45.55±0.65 | 55.29±0.59 | 29.63±0.17 | 44.68±0.17 | 29.62±0.64 | 36.25±0.14 | 30.28±0.52 | 35.63±0.52 |
| + SATCH (ours) | **52.09±0.68** | **58.48±0.32** | **37.48±0.51** | **47.07±0.28** | **36.75±0.46** | **38.84±0.94** | **32.79±0.42** | **37.83±0.58** |

Note: − denotes results that were incompatible with the GCIL setting.

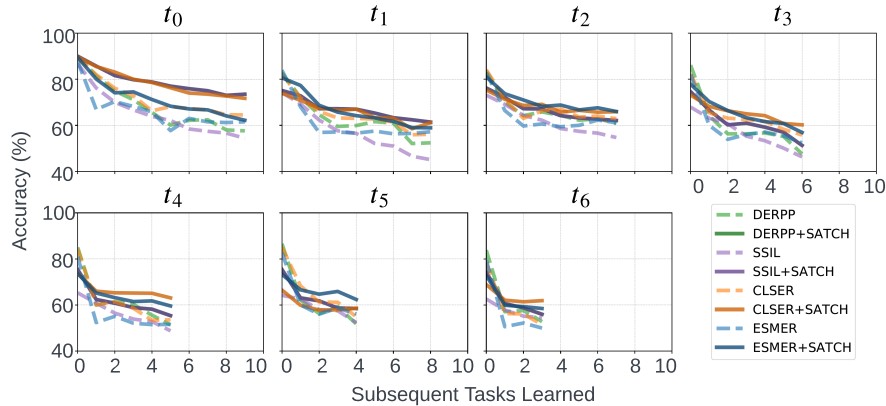

Figure 3: Accuracy of individual tasks as subsequent tasks are learned on CIFAR100 with buffer size 5000. i.e. if $t = t_1$ and there are 10 tasks, then 8 subsequent tasks can be learned.

**Ablation Study**: To assess each component in SATCH, we conducted an ablation study shown in Table 2. The key components incorporating the assistant teacher's knowledge: new task learning (NEWL), which distills soft labels from the assistant teacher to guide the student; diverse knowledge (DIVK), which combines the assistant and main teacher's logits for complementary knowledge transfer; and buffer selection (BUFS), which refines sample selection based on the agreement between the student and assistant teacher. To ensure all components are evaluated equally, we include task-recency bias mitigation (BIASM), which reduces forgetting of past tasks by distilling only the current task's logits during training. Each component improves accuracy, either individually or in combination. For instance, when BUFS is removed, we observe a drop in accuracy across all methods, indicating that this component also improves accuracy in non-noisy environments.

**Effect of Different Teacher Backbones:** We evaluate the impact of varying SATCH's assistant teacher backbone, focusing on how different backbones affect accuracy for our NEWL, DIVK, and BUFS components. We compare three backbones: $SATCH_{lg}$, which uses the same ResNet-18 backbone as the student model; $SATCH_{sm}$, a reduced version of ResNet-18 (Kang et al., 2022); and $SATCH_{conv}$, a compact 3-layer convolutional network (Ramesh & Chaudhari, 2022). Table 3

Table 2: Ablation study of SATCH on CIFAR100 with buffer size 5000. Accuracy and standard deviation over five runs.

| BIAM | NEWL | DIVK | BUFS | DER++ + SATCH | SSIL + SATCH | CLSER + SATCH | ESMER + SATCH |
|---|---|---|---|---|---|---|---|
| ✓ | ✓ | ✓ | ✓ | 59.97±0.18 | 54.06±0.64 | 62.45±0.39 | 58.48±0.32 |
| ✓ | ✓ | ✓ | ✗ | 59.85±0.38 | 53.52±0.38 | 62.12±0.63 | 58.22±0.09 |
| ✓ | ✗ | ✓ | ✗ | 59.67±0.06 | 53.25±0.62 | 61.61±0.45 | 57.43±0.65 |
| ✓ | ✓ | ✗ | ✗ | 58.83±0.78 | 52.07±0.50 | 61.58±0.61 | 57.22±0.30 |
| ✓ | ✗ | ✗ | ✗ | 58.67±0.11 | 51.54±0.89 | 60.70±0.33 | 56.11±0.37 |
| ✗ | ✗ | ✗ | ✗ | 56.39±1.06 | 51.54±0.89 | 59.63±1.12 | 55.29±0.59 |

shows that the backbone used impacts accuracy. With the smallest model, $\text{SATCH}_{conv}$, there is only a 0.09% improvement when using NEWL, likely due to the capacity gap between the student and teacher (Son et al., 2021). However, using DIVK results in a 1.03% improvement compared to 0.41% for $\text{SATCH}_{lg}$, which has significantly more parameters, suggesting the improvement may be due to architectural diversity (You et al., 2017). $\text{SATCH}_{sm}$ achieves the highest accuracy for both NEWL and DIVK with improvements of 1.11% and 1.31%, respectively, indicating that the backbone needs to strike a balance between minimizing the capacity gap for NEWL and providing enough architectural diversity for DIVK. Additionally, incorporating specialized knowledge with the same backbone as the student in $\text{SATCH}_{lg}$ improves new task learning by 1.02%, suggesting that specialized knowledge can improve student learning even when the same backbone is used.

Table 3: Ablation study with different assistant teacher backbones in SATCH on CIFAR100 with buffer size 5000. Accuracy and standard deviation over five runs.

| BIAM | NEWL | DIVK | BUFS | ESMER + $\text{SATCH}_{lg}$ | ESMER + $\text{SATCH}_{sm}$ | ESMER + $\text{SATCH}_{conv}$ |
|---|---|---|---|---|---|---|
| ✓ | ✓ | ✓ | ✓ | 57.65±0.31 | 58.48±0.32 | 55.25±0.42 |
| ✓ | ✓ | ✓ | ✗ | 57.60±0.14 | 58.22±0.09 | 56.69±0.73 |
| ✓ | ✗ | ✓ | ✗ | 56.52±0.47 | 57.43±0.65 | 57.14±0.62 |
| ✓ | ✓ | ✗ | ✗ | 57.13±0.43 | 57.22±0.30 | 56.20±0.27 |
| ✓ | ✗ | ✗ | ✗ | 56.11±0.37 | 56.11±0.37 | 56.11±0.37 |
| ✗ | ✗ | ✗ | ✗ | 55.29±0.59 | 55.29±0.59 | 55.29±0.59 |

**Visualizations of SATCH**: In our study, complementary knowledge refers to broader and more generalized feature representations that allow the student model to retain prior knowledge while reducing overfitting to specific tasks. To evaluate the diversity of knowledge introduced by SATCH, we compare the Grad-CAM++ visualizations generated by the assistant teacher in SATCH and the main teacher, as shown in Figure 4. The feature map generated by the assistant teacher $\phi^t$ in $\text{SATCH}_{sm}$ focuses on different features compared to the main teacher in ESMER which emphasizes general and background features. When combined, these complementary focuses result in a broader and more comprehensive feature map. For example, in the case of the beaver image, the assistant teacher highlights the main body of the beaver, while ESMER captures additional background details. The combined model consequently emphasizes a larger and more detailed portion of the beaver's body, suggesting that SATCH complements the main teacher by capturing specialized features.

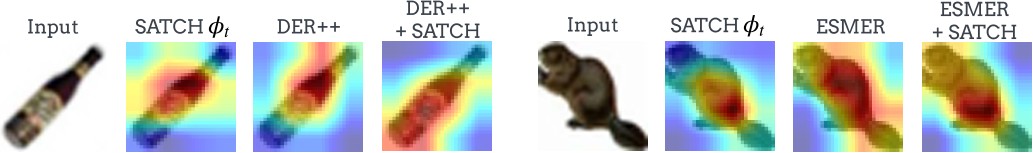

Figure 4: Grad-CAM++ (Chattopadhay et al., 2018) visualization between the features of $\text{SATCH}_{sm}$ assistant teacher $\phi^t$, DER++, DER++ + SATCH, ESMER, and ESMER + SATCH.

**Noisy Labels in a Real World Scenario:** We evaluate SATCH's ability to filter out noisy labels, as storing these samples can cause models to memorize incorrect labels (Sarfraz et al., 2023; Kim et al., 2021). Table 4 demonstrates the accuracy of various methods under different levels of label noise. SATCH consistently outperforms baseline methods by using the assistant teacher's complementary knowledge to reduce the impact of noisy labels and reduce catastrophic forgetting. We integrate SATCH with ESMER, which uses an error-sensitive reservoir sampling technique that filters noisy samples during learning. ESMER's buffer selection is modified only to include samples where both SATCH and ESMER agree, adding an extra layer of noise detection (ablation study under noisy setting in Appendix A.7). When ESMER is combined with SATCH, accuracy is increased by 3.86%, 7.61%, and 7.71% on CIFAR100 under 10%, 25%, and 50% of label noise, respectively. These results demonstrate that SATCH effectively reduces the negative impact of noisy labels.

Table 4: Effect of varying degrees of label noise on CIFAR100 and TinyImageNet datasets with a buffer size of 5000. Accuracy and standard deviation over five runs.

| | CIFAR100 | | | TinyImageNet | | |
|---|---|---|---|---|---|---|
| Label Noise | 10% | 25% | 50% | 10% | 25% | 50% |
| JOINT (Upper bound) | 62.86±0.41 | 59.09±0.27 | 51.62±0.39 | 50.62±0.19 | 45.82±0.51 | 40.09±0.26 |
| SGD (Lower bound) | 7.44±0.38 | 6.52±0.40 | 5.31±0.33 | 6.66±0.18 | 5.51±0.11 | 3.42±0.04 |
| ER (Neurips 19) | 28.46±0.77 | 21.34±1.23 | 11.30±0.72 | 18.28±0.58 | 11.50±0.32 | 5.55±0.06 |
| ER-ACE (ICLR 22) | 44.56±0.62 | 31.11±1.28 | 16.16±0.55 | 26.08±0.24 | 17.27±0.26 | 7.78±0.29 |
| DER++ (Neurips 20) | 44.83±0.45 | 31.22±1.08 | 16.88±0.42 | 24.32±0.55 | 16.00±0.11 | 7.68±0.10 |
| + SATCH (ours) | **51.04±0.68** | **41.88±0.17** | **23.68±0.46** | **35.28±0.72** | **25.75±0.49** | **13.02±0.73** |
| SSIL (ICCV 21) | 42.15±0.30 | 32.28±0.72 | 19.09±0.38 | 32.45±0.91 | 23.62±0.34 | 12.25±0.36 |
| + SATCH (ours) | **45.83±0.85** | **39.90±0.61** | **26.96±0.71** | **34.79±0.55** | **26.52±0.17** | **15.63±0.35** |
| CLS-ER (ICLR 22) | 48.37±0.46 | 35.59±0.55 | 17.46±0.46 | 29.60±0.40 | 19.18±0.33 | 9.11±0.38 |
| + SATCH (ours) | **53.73±0.19** | **43.21±0.54** | **25.81±0.52** | **38.33±0.25** | **28.19±0.30** | **13.99±0.25** |
| ESMER (ICLR 23) | 48.50±0.64 | 37.01±0.52 | 20.82±0.33 | 36.77±0.69 | 27.43±0.85 | 13.49±0.93 |
| + SATCH (ours) | **52.36±0.18** | **44.62±0.39** | **28.53±0.46** | **40.49±0.11** | **32.46±0.13** | **18.96±0.29** |

**Effect on Task-Specific Knowledge**: SATCH maintains task-specific knowledge over time, which is forgotten by the main teacher of knowledge distillation methods when learning multiple tasks. We get the task accuracy over time for existing knowledge distillation methods by using the task label to select the subset of output logits from the single head. Figure 5 (a) shows that most knowledge distillation methods lose task-specific accuracy as more tasks are learned, except SSIL, as it maintains a separated softmax that prevents interference when learning other tasks. Figure 5 (b) compares the accuracy for each task after all tasks have been learned. The accuracy for other methods is generally lower than SATCH apart from the last task, which may be due to task-recency bias.

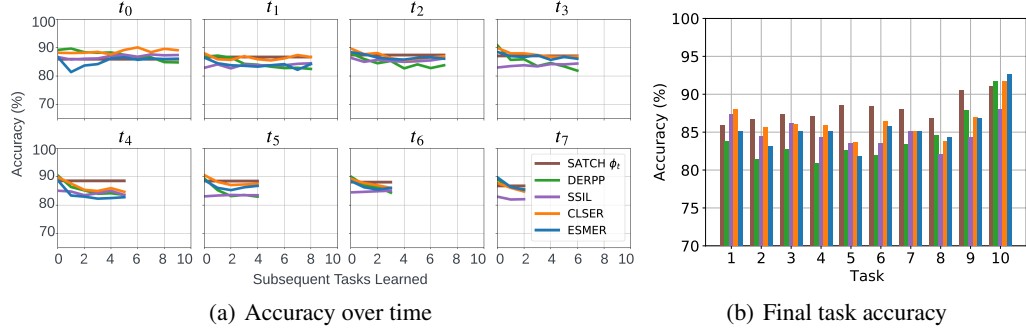

(a) Accuracy over time      (b) Final task accuracy

Figure 5: Accuracy given the task ID on CIFAR100 with 5000 buffer size. (a) Task accuracy for specific tasks as more tasks are learned. (b) Final task accuracy after all tasks have been learned.

## 5 CONCLUSION

In this paper, we introduced SATCH, a novel continual learning framework that incorporates a specialized assistant teacher to reduce catastrophic forgetting. The assistant teacher, using a smaller backbone, is trained exclusively on the current task and offers complementary knowledge while preserving task-specific information. SATCH guides new task learning, refines buffer selection, and effectively combines the assistant teacher's specialized knowledge with the main teacher's generalized knowledge, striking a balance between retaining old knowledge and learning new knowledge. Our experimental results demonstrate that SATCH achieves state-of-the-art accuracy when integrated with existing knowledge distillation methods, improving both accuracy and robustness, particularly in noisy environments. Ablation studies confirm that each component of SATCH is essential for these improvements in accuracy.

**Reproducibility Statement** Several steps have been taken to ensure that the results presented in this paper are reproducible. We provide a comprehensive codebase from which all the results were generated in the supplementary materials, including the optimal hyperparameters for each method and learning setting. A README file is also included to guide users in reproducing our results. Details of all hyperparameters are clearly outlined both in the appendix and codebase.

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

# A   APPENDIX

---

**Algorithm 1** Specialized Assistant Teacher Distillation (SATCH) Algorithm

---

**Require:** Assistant teacher $\phi_t$; main teacher $\theta_{t:t-1}^{mt}$; student model $\theta_{1:t}^s$; episodic memory $\mathcal{M}$; task-specific mask $m_t$; guiding distillation weight $\lambda$

1: Initialize: $\mathcal{M} \leftarrow \{\}$
2: **while** Get a mini-batch of samples $(x_t, y_t) \sim \mathcal{D}_t$ **do**
3:     Sample memory $(x_t^m, y_t^m, z_t^{\phi_t}) \sim \mathcal{M}$
4:     Get student model logits: $z_{1:t}^s = \theta_{1:t}^s(x_t)$
5:     Apply task-specific mask to get task $t$ logits: $z_t^s = m_t \odot z_{1:t}^s$
6:     Get assistant teacher logits: $z_t^{\phi_t} = \phi_t(x_t)$
7:     Compute $\mathcal{L}_{NT}$ using cross-entropy and KL divergence (Eq. 2)
8:     Sample buffer, get main teacher logits: $z_t^{mt} = \theta_m(x_t^m)$
9:     Compute masked average logits of assistant and main teacher: $\hat{z}_t^{mt} = \frac{1}{2}(z_t^{mt} + z_t^{\phi_t})$ (Eq. 4)
10:     Combine with main teacher logits: $\hat{z}_{1:t}^{mt} = \bar{m}_t \odot z_{1:t}^{mt} + \hat{z}_t^{mt}$ (Eq. 5)
11:     Compute $\mathcal{L}_{OT}$ using cross-entropy and KL divergence (Eq. 6)
12:     Combine overall loss: $\mathcal{L}_{total} = \mathcal{L}_{NT} + \mathcal{L}_{OT}$
13:     Filter and select representative samples: $(x_c, y_c) \leftarrow \{(x_i, y_i) \mid \hat{y}_{s,i} = \hat{y}_{a,i}\}$ (Eq. 3)
14:     Update buffer with assistant teacher logits: $\mathcal{M} \leftarrow \text{Reservior}(\mathcal{M}, (x_c, y_c, z_t^{\phi_t}))$
15: **end while**
16: **return** student model

---

## A.1   IMPLEMENTATION DETAILS

We use ResNet-18 network and train with SGD optimizer. For all our experiments, we train for 50 epochs with a batch size of 32 for both incoming samples and memory and apply random crop and horizontal flip data augmentations. Following DER++ (Buzzega et al., 2020), we store non-augmented images in the buffer and apply the data augmentations to the memory samples for replay. We trained our models on an Nvidia Tesla A100 GPU with 40GB of memory.

## A.2   PERFORMANCE METRICS

Following Lin et al. (2022), we use accuracy and forgetting to evaluate performance. Accuracy it is the average test classification accuracy of all tasks and forgetting which measures the accuracy difference of old tasks after learning new tasks. Formally, accuracy and forgetting are defined as:

$$\text{accuracy} = \frac{1}{T}\sum_{i=1}^{T} A_{T,i}, \text{forgetting} = \frac{1}{T-1}\sum_{i=1}^{T-1} A_{T,i} - A_{i,i} \tag{7}$$

Here, $T$ is the total number of sequential tasks and $A_{T,i}$ is the accuracy of the model on $i$-th task after learning the $T$-th task sequentially (Lopez-Paz & Ranzato, 2017).

## A.3   BASELINES

We evaluate our approach by focusing on replay-based and knowledge distillation methods, as these have been shown to outperform other techniques in the class incremental learning setting (Chaudhry et al., 2019; Arani et al., 2022). The following baselines are considered:

**JOINT**: provides the upper bound by jointly learning all tasks.

**SGD**: provides the lower bound by fine-tuning the model when a new task is learned.

**ER**: Chaudhry et al. (2019) performs interleaved training of the new task and the memory sample to approximate the joint distribution of all tasks.

**ER-ACE**: Caccia et al. (2022) addresses the issue of task transition causing a significant drift in the learned representations of previous classes by introducing an asymmetric cross-entropy loss that only considers the logits of new classes.

**DER++**: Buzzega et al. (2020) introduces an additional distillation loss by storing output logits alongside samples and enforcing consistency in the output space.

**PASS**: Zhu et al. (2021) uses prototypes, which act as anchors for each class in feature space. It incorporates self-supervised auxiliary tasks, such as contrastive learning, to improve feature robustness.

**SS-IL**: Ahn et al. (2021) learns the current task loss and replay loss in isolation from each other, using task-specific distillation on the rehearsal data.

**CLS-ER**: Arani et al. (2022) models the interaction between fast and slow learning systems by maintaining two semantic memories that update the model's weights at different rates using an exponential moving average.

**ESMER**: Sarfraz et al. (2023) proposes that the model should learn more from samples with smaller losses to avoid large feature drift, assigning different weights to new samples based on their loss value.

### A.4 DATASETS

#### A.4.1 CLASS INCREMENTAL LEARNING

Class-incremental learning evaluates continual learning in a more realistic scenario where the task ID is unavailable during inference. We use the CIFAR100, TinyImageNet, and MiniImageNet datasets, with the classes split into 10 tasks containing disjoint sets of 10, 20, and 10 classes, respectively. In this setting, all methods are evaluated without using separate classification heads, as the task ID is not provided during inference.

#### A.4.2 GENERALIZED CLASS INCREMENTAL LEARNING

Generalized class-incremental learning (Mi et al., 2020) evaluates continual learning in a more realistic scenario where tasks may have overlapping classes, the number of classes varies across tasks, training instances per task are inconsistent during inference.

#### A.4.3 NOISY CLASS INCREMENTAL LEARNING

Noisy class incremental learning evaluates the robustness of a continual learning model when faced with label noise. In real-world data streams, noise is inevitable (Sarfraz et al., 2023), and an effective continual learning method must handle both noisy labels and incremental learning, avoiding memorization of noise and forgetting. We apply varying degrees of symmetric label noise to the CIFAR100 and TinyImageNet datasets. Following Sarfraz et al. (2023), for each task, we sample random labels from the classes in the task and replace a fraction of the original labels with the random ones. Note that, under this setting, the random label may correspond to the original label.

### A.5 RUNTIME AND MEMORY CONSUMPTION

We evaluated the memory and runtime efficiency of SATCH when integrated with SSIL during the learning of the last task. We capture the memory and runtime performance while learning a new task. We measured memory usage (in MB) and runtime (in epochs per hour, ep/hour) as shown in Table 5. We compare SATCH with ANCL (Kim et al., 2023), an auxiliary method that pre-trains a copy of the student to learn the new task and distill task-specific knowledge. Auxiliary teacher methods (Kim et al., 2023) involve of two key steps: (1) training the assistant teacher before the student and (2) using the assistant teacher to modify the knowledge transferred during knowledge distillation. For step (1), the memory usage of SATCH$_{lg}$ and ANCL is similar, as both use a ResNet-18 backbone. However, ANCL requires approximately twice the runtime, as it also performs replay and distillation to retain previous knowledge, while SATCH focuses only on the current task. SATCH$_{sm}$, which utilizes a reduced ResNet-18 backbone, reduces memory consumption by 26% and runs approximately 3.5 times faster than ANCL. To isolate the effects of the additional runtime and memory during the distillation process in step (2), we excluded the pre-training of the assistant teacher. SATCH incurs

a slightly higher memory cost due to storing specialized logits alongside replay samples. ANCL's lower ep/hour is due to the additional regularization term during learning, whereas SATCH's reduced ep/hour results from using the assistant teacher's knowledge in more ways during training.

Table 5: Memory and runtime of SATCH on CIFAR100 with 5000 buffer size on the last task

|  | Memory (MB) | Runtime (ep/hour) |
| --- | --- | --- |
| SATCH$_{conv}$ Assistant Teacher | 26 | 1759 |
| SATCH$_{sm}$ Assistant Teacher | 73 | 856 |
| SATCH$_{lg}$ Assistant Teacher | 276 | 770 |
| SSIL++ + ANCL Teacher | 282 | 342 |
| SSIL++ (ICCV 21) | 1012 | 380 |
| + ANCL (CVPR 23) | 1012 | 303 |
| + SATCH$_{sm}$ (ours) | 1016 | 294 |

## A.6 Comparison with ANCL

We compare SATCH with ANCL (Kim et al., 2023), the most similar method to ours. ANCL proposes a framework where an auxiliary network is trained on the current task and used as an additional regularization term during student network training. In traditional self-distillation methods, a copy of the student network is treated as a teacher model, $\theta_{1:t-1}^t$, which retains knowledge from previous tasks. ANCL introduces an additional copy of the student, designed to learn the new task before guiding the student network with old and new knowledge. While ANCL does guide new tasks, it does not capture task-specific knowledge or introduce backbone diversity. Furthermore, ANCL requires more runtime and memory than SATCH, as the auxiliary teacher must replicate the original method's learning process to prevent forgetting. In contrast, SATCH's assistant teacher only specializes in the current task.

We evaluate SATCH and ANCL in Table 6 when combined with DER++ and SSIL. To ensure a fair comparison with ANCL on DER++, we apply task recency bias mitigation and only evaluate the components that use the assistant teacher knowledge. Although ANCL shows accuracy improvements, SATCH has higher accuracy than ANCL when combined with DER++ and SSIL. These results indicate that SATCH can effectively utilize specialized teacher knowledge, whereas ANCL relies on pre-training a copy of the student for the new task.

Additionally, SATCH improves runtime and memory efficiency, as shown in Table 5. ANCL's need to pre-train a student copy increases its runtime by incorporating the base method's distillation and replay process. At the same time, SATCH's assistant teacher focuses exclusively on learning the current task without the need to perform replay or knowledge distillation. In conclusion, SATCH's assistant teacher provides task-specific knowledge that reduces forgetting by introducing complementary information with improved runtime and lower memory over ANCL.

Table 6: Accuracy and standard deviation over five different runs per dataset on ResNet-18 with Task-Recency Bias Mitigation (BIAM) with 5000 buffer size.

|  | CIFAR100 | TinyImageNet |
| --- | --- | --- |
| DER++ (Neurips 20) | $56.39_{\pm1.06}$ | $36.39_{\pm0.45}$ |
| + ANCL (CVPR 23) w/ BIAM | $57.99_{\pm0.38}$ | $40.15_{\pm0.29}$ |
| + SATCH (ours) | $\mathbf{59.97_{\pm0.18}}$ | $\mathbf{42.33_{\pm0.26}}$ |
| SSIL (ICCV 21) | $51.54_{\pm0.89}$ | $39.13_{\pm0.47}$ |
| + ANCL (CVPR 23) | $51.56_{\pm0.85}$ | $40.73_{\pm0.61}$ |
| + SATCH (ours) | $\mathbf{54.06_{\pm0.64}}$ | $\mathbf{42.51_{\pm0.27}}$ |

## A.7 Ablation study for noisy continual learning

We perform an ablation study on CIFAR100 under a noisy label setting, where 50% of the labels are randomly corrupted with 5000 buffer size as shown in Table 11. The key components that incorporate the assistant teacher's knowledge include: new task learning (NEWL), which distills soft labels from the assistant teacher to guide the student; diverse knowledge (DIVK), which combines the assistant and main teacher's logits for more complementary knowledge transfer; and buffer selection (BUFS), which refines sample selection based on the agreement between the student and assistant teacher. Additionally, to ensure all components are evaluated equally, we include task-recency bias mitigation (BIASM), which reduces forgetting of past tasks by distilling only the current task's logits during training. Each component contributes to overall accuracy. Adding NEWL yields a further gain of 0.69%. The combination of BIAM, NEWL, and DIVK leads a combined accuracy boost of 2.16%, demonstrating the complementary nature of SATCH's components. Incorporating BUFS alongside BIAM, NEWL, and DIVK increases the accuracy by 4.55%, highlighting the important role of BUFS in refining the buffer by selecting samples where the student and assistant teacher agree, thus improving the model's robustness to label noise.

Table 7: Ablation study of ESMER + SATCH on Noisy-CIFAR100 with buffer size 5000 with 50% label corruption. Accuracy and standard deviation over five runs.

| BIAM | NEWL | DIVK | BUFS | 25% | 50% |
|:---:|:---:|:---:|:---:|:---:|:---:|
| ✓ | ✓ | ✓ | ✓ | $44.62_{\pm0.39}$ | $28.53_{\pm0.46}$ |
| ✓ | ✓ | ✓ | ✗ | $43.79_{\pm0.60}$ | $26.14_{\pm0.16}$ |
| ✓ | ✗ | ✓ | ✗ | $42.20_{\pm0.13}$ | $24.67_{\pm0.37}$ |
| ✓ | ✓ | ✗ | ✗ | $43.26_{\pm0.63}$ | $25.53_{\pm0.20}$ |
| ✓ | ✗ | ✗ | ✗ | $41.71_{\pm0.53}$ | $23.98_{\pm0.11}$ |
| ✗ | ✗ | ✗ | ✗ | $37.01_{\pm0.52}$ | $20.82_{\pm0.33}$ |

## A.8 Noisy Labels Stored in the Buffer

SATCH uses an additional assistant teacher to identify noisy labels using different knowledge from the main teacher and our assistant teacher. This reduces overfitting on noisy labels and improves learning stability. The model must handle incremental learning and label noise in the Noisy-Class-IL setting. SATCH's ability to refine the buffer selection is important in noisy environments where replaying mislabeled samples can further increase forgetting.

We investigate the percentage of noisy labels in the buffer when combining SATCH with ES-MER (Sarfraz et al., 2023). ESMER incorporates an error-sensitive reservoir sampling component to filter noisy samples during learning. By combining this with SATCH's buffer selection, we show further improvements in accuracy for noisy continual learning. Both SATCH and ESMER's buffer selection techniques must agree that a sample is not noisy before storing it. We compare the percentage of noisy labels stored in the buffer using ESMER and SATCH. Figure 6 shows the percentage of noisy samples stored in the buffer across ER-ACE, ESMER, and ESMER+SATCH. While ER-ACE retains approximately 50% of corrupted labels due to random sampling, ESMER's error-sensitive reservoir sampling significantly reduces this. SATCH's teacher-student agreement further filters out noisy samples missed by ESMER, demonstrating its effectiveness in minimizing the retention of corrupted labels.

## A.9 Hyperparameter Tuning

SATCH introduces a $\lambda$ hyperparameter to control the guidance from the assistant teacher, transferring task-specific knowledge to the student. The $\lambda$ hyperparameter is selected among four different values [0.1, 0.4, 0.7, 1] based on validation sets split from the training data on CIFAR100 with 5000 buffer size. Our findings show that lower values of $\lambda$, generally lead to higher accuracy. Setting $\lambda$ too high can lead to overfitting on the teacher model's knowledge. Previous work has explored ways to prevent overfitting, such as dynamically adjusting the weighting throughout training (Lee et al., 2019; Hinton, 2015). Based on this insight, we set $\lambda = 0.1$ for all buffer sizes and datasets to reduce the method's dependency on extensive hyperparameter tuning.

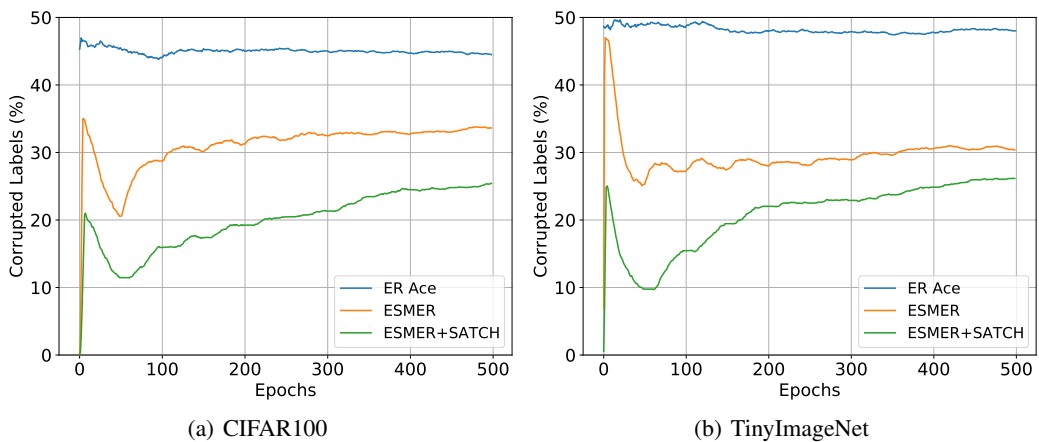

(a) CIFAR100                  (b) TinyImageNet

Figure 6: Noisy samples stored over time for ER-ACE, ESMER, and ESMER+SATCH on CI-FAR100 with buffer size 5000

It is important to note that both DER++, SSIL, CLS-ER, and ESMER have their specific hyperparameters that may affect the accuracy. To ensure a fair comparison, we keep the original hyperparameters of each method consistent when combined with SATCH. This means that we use the same hyperparameter settings as the original implementation for evaluation. Furthermore, we follow the mammoth continual learning repository (Buzzega et al., 2020) to set the specific hyperparameters in each method or as reported in the respective code repository. Since our experimental setup aligns with prior work, hyperparameters such as the learning rate remain consistent with the original code. When combining SATCH with other knowledge distillation methods, all hyperparameters are kept consistent with the original methods, including the learning rate and batch sizes. The only hyperparameter added for our method is the use of $\lambda$ that controls the impact of the new task guidance when from SATCH's assistant teacher.

A.10   FORGETTING MEASURE

We evaluate the average forgetting of SATCH on various knowledge distillation methods in Table 9 and Table 8.

Table 8: Comparison of CL methods with varying datasets and buffer sizes. We report average forgetting and standard deviation over 3 different runs for each result.

|  | CIFAR100 | | TinyImageNet | | MiniImageNet | |
| --- | --- | --- | --- | --- | --- | --- |
| Memory Size | 1000 | 5000 | 1000 | 5000 | 1000 | 5000 |
| ER (Neurips 19) | $46.25_{\pm2.32}$ | $28.15_{\pm1.48}$ | $52.94_{\pm0.63}$ | $34.90_{\pm0.29}$ | $46.60_{\pm1.67}$ | $30.89_{\pm1.71}$ |
| ER-ACE (ICLR 22) | $25.42_{\pm1.44}$ | $14.31_{\pm0.26}$ | $30.73_{\pm0.68}$ | $21.22_{\pm0.81}$ | $25.67_{\pm0.92}$ | $21.31_{\pm1.10}$ |
| DER++ (Neurips 20) | $32.88_{\pm0.49}$ | $16.01_{\pm0.82}$ | $52.71_{\pm0.90}$ | $29.79_{\pm0.76}$ | $48.05_{\pm1.03}$ | $36.54_{\pm0.83}$ |
| + SATCH (ours) | $22.34_{\pm0.23}$ | $9.73_{\pm0.42}$ | $30.70_{\pm0.71}$ | $15.16_{\pm0.49}$ | $23.77_{\pm1.03}$ | $12.46_{\pm0.67}$ |
| SSIL (ICCV 21) | $19.36_{\pm0.24}$ | $15.07_{\pm0.15}$ | $20.16_{\pm0.42}$ | $14.91_{\pm0.41}$ | $11.58_{\pm0.86}$ | $15.21_{\pm0.92}$ |
| + SATCH (ours) | $17.30_{\pm0.86}$ | $13.57_{\pm0.45}$ | $17.02_{\pm0.67}$ | $10.41_{\pm0.29}$ | $8.18_{\pm1.06}$ | $11.54_{\pm0.52}$ |
| CLS-ER (ICLR 22) | $29.31_{\pm0.76}$ | $13.79_{\pm0.01}$ | $46.71_{\pm1.10}$ | $27.65_{\pm0.95}$ | $40.83_{\pm1.48}$ | $33.03_{\pm1.54}$ |
| + SATCH (ours) | $14.86_{\pm0.19}$ | $10.07_{\pm0.76}$ | $17.26_{\pm0.57}$ | $16.14_{\pm0.66}$ | $10.03_{\pm0.22}$ | $23.80_{\pm0.87}$ |
| ESMER (ICLR 23) | $29.81_{\pm1.13}$ | $13.82_{\pm1.66}$ | $43.10_{\pm0.71}$ | $27.02_{\pm0.22}$ | $37.01_{\pm1.48}$ | $26.69_{\pm0.72}$ |
| + SATCH (ours) | $18.08_{\pm0.97}$ | $12.10_{\pm0.93}$ | $20.24_{\pm1.02}$ | $14.96_{\pm0.48}$ | $24.85_{\pm0.38}$ | $21.71_{\pm1.11}$ |

Table 9:  Effect of varying degrees of label noise on CIFAR100 and TinyImageNet datasets with a buffer size of 5000. Average forgetting and standard deviation over 3 runs for each result.

| Label Noise | CIFAR100 | | | TinyImageNet | | |
|---|---|---|---|---|---|---|
| | 10% | 25% | 50% | 10% | 25% | 50% |
| ER (Neurips 19) | $38.57_{\pm1.69}$ | $38.30_{\pm1.52}$ | $31.22_{\pm0.91}$ | $36.98_{\pm1.07}$ | $35.41_{\pm1.01}$ | $25.79_{\pm0.48}$ |
| ER-ACE (ICLR 22) | $18.87_{\pm1.02}$ | $19.89_{\pm1.08}$ | $16.15_{\pm0.65}$ | $26.03_{\pm0.49}$ | $24.89_{\pm0.54}$ | $17.60_{\pm0.42}$ |
| DER++ (Neurips 20) | $27.16_{\pm0.70}$ | $34.90_{\pm1.43}$ | $33.29_{\pm0.52}$ | $38.20_{\pm0.42}$ | $40.62_{\pm0.45}$ | $33.77_{\pm0.56}$ |
| + SATCH (ours) | $12.91_{\pm0.34}$ | $9.43_{\pm0.62}$ | $10.03_{\pm0.55}$ | $18.59_{\pm0.29}$ | $21.33_{\pm0.22}$ | $16.36_{\pm0.68}$ |
| SSIL (ICCV 21) | $12.77_{\pm0.38}$ | $13.33_{\pm0.29}$ | $10.98_{\pm0.26}$ | $14.11_{\pm0.56}$ | $15.23_{\pm0.44}$ | $11.79_{\pm0.18}$ |
| + SATCH (ours) | $9.00_{\pm1.10}$ | $10.39_{\pm0.62}$ | $10.73_{\pm0.68}$ | $9.34_{\pm0.31}$ | $11.69_{\pm0.60}$ | $10.71_{\pm0.59}$ |
| CLS-ER (ICLR 22) | $21.56_{\pm0.29}$ | $26.52_{\pm0.74}$ | $29.30_{\pm1.13}$ | $30.67_{\pm0.33}$ | $34.05_{\pm0.25}$ | $26.80_{\pm0.65}$ |
| + SATCH (ours) | $8.36_{\pm0.40}$ | $8.69_{\pm0.82}$ | $8.35_{\pm0.70}$ | $13.61_{\pm0.44}$ | $16.87_{\pm0.27}$ | $15.63_{\pm0.44}$ |
| ESMER (ICLR 23) | $15.99_{\pm1.34}$ | $15.66_{\pm0.63}$ | $14.39_{\pm0.17}$ | $24.97_{\pm0.59}$ | $25.69_{\pm1.36}$ | $21.60_{\pm0.33}$ |
| + SATCH (ours) | $8.48_{\pm0.12}$ | $6.97_{\pm0.31}$ | $6.00_{\pm0.55}$ | $12.34_{\pm0.58}$ | $12.54_{\pm0.29}$ | $10.36_{\pm0.14}$ |

Table 10:  Comparison of backbone networks with equal parameter sizes.

| Memory Size | CIFAR100 | |
|---|---|---|
| | 1000 | 5000 |
| DER++$_{enlarged}$ | $45.49_{\pm0.63}$ | $57.80_{\pm0.73}$ |
| DER++ | $44.62_{\pm0.56}$ | $56.39_{\pm1.06}$ |
| + SATCH (ours) | $\mathbf{48.38_{\pm0.19}}$ | $\mathbf{59.97_{\pm0.18}}$ |
| SSIL$_{enlarged}$ | $41.17_{\pm0.49}$ | $52.61_{\pm1.01}$ |
| SSIL | $40.70_{\pm0.40}$ | $51.54_{\pm0.89}$ |
| + SATCH (ours) | $\mathbf{42.95_{\pm0.17}}$ | $\mathbf{54.06_{\pm0.64}}$ |
| CLS-ER$_{enlarged}$ | $47.44_{\pm0.59}$ | $60.02_{\pm0.95}$ |
| CLS-ER | $45.47_{\pm0.63}$ | $59.63_{\pm1.12}$ |
| + SATCH (ours) | $\mathbf{52.36_{\pm0.30}}$ | $\mathbf{61.39_{\pm0.30}}$ |
| ESMER$_{enlarged}$ | $45.95_{\pm0.49}$ | $56.82_{\pm0.64}$ |
| ESMER | $45.55_{\pm0.65}$ | $55.29_{\pm0.59}$ |
| + SATCH (ours) | $\mathbf{52.09_{\pm0.68}}$ | $\mathbf{58.48_{\pm0.32}}$ |

Table 11:  Total trainable parameter count of backbone models used.

| Backbone | Parameter Count |
|---|---|
| Enlarged (Resnet + Reduced Resnet) | 12.85M |
| Baseline (Resnet) | 11.40M |
| SATCH$_{sm}$ (Reduced Resnet) | 1.25M |
| SATCH$_{conv}$ (Simple 3 conv layer) | 0.69M |

