# OpenReview forum: "SATCH: Specialized Assistant Teacher Distillation to Reduce Catastrophic Forgetting"
_ICLR.cc/2025/Conference — Submitted to ICLR 2025_

### Official Review · Reviewer_kU7R · 2024-11-02

**Soundness:** 1
**Presentation:** 1
**Contribution:** 1
**Rating:** 5
**Confidence:** 4

**Summary:**

This paper introduces a method for continual learning that addresses key challenges of existing knowledge distillation based class-incremental strategies used in them. Traditional methods often struggle with the loss of task-specific knowledge, limited diversity in knowledge transfer, and delays in teacher model availability. SATCH proposes the use of a smaller assistant teacher trained on the current task to offer task-specific guidance early in the learning process. This approach diversifies and enhances the knowledge transferred to the student model while refining sample selection in noisy environments. Experimental results on standard continual learning benchmarks, such as CIFAR100, TinyImageNet, and MiniImageNet, show that SATCH improves accuracy by up to 12% compared to state-of-the-art methods. The paper highlights SATCH’s robust integration with existing frameworks and emphasizes its contributions to mitigating catastrophic forgetting through improved knowledge diversity and task-specific retention​.

**Strengths:**

1. Improved Knowledge Diversity: By combining the specialized knowledge of the assistant teacher with the generalized knowledge of the main teacher, SATCH effectively diversifies the knowledge transfer process. This approach enriches the learning experience for the student model and mitigates the limitations of using a single teacher model.

2. Integration with Existing Methods: The method is designed to work seamlessly with established distillation based class-incremental learning methods

**Weaknesses:**

1. Limited Discussion on Computational Overheads: The assistant teacher’s additional computations may raise concerns for resource-constrained environments and by makes existing methods computationally inefficient. In additition, the assistant teacher training followed by the distillation performed, makes the knowledge transfer process cumbersome. Adding a detailed analysis of the computational complexity and runtime of SATCH compared to baseline methods. Quantifying the impact on memory and processing time across various settings would clarify the scalability of the approach. Additionally, consider exploring potential optimizations to make the process more efficient, such as parallel training strategies, etc.

2. Lack of Broader Comparisons: The contributions in the paper are limited to a particular kind class-incremental paradigm, therefore its applicability in a broader context remains limited. The paper could also have strengthened its argument by comparing SATCH against a wider variety of lifelong learning or parameter isolation methods. This omission weakens the case for its effectiveness. To strengthen the argument for SATCH’s effectiveness, The authors could expand the comparative study to include more diverse continual learning approaches, such as parameter isolation techniques (e.g., Progressive Neural Networks or Elastic Weight Consolidation). This would help assess the general applicability and robustness of SATCH across various scenarios. Furthermore, a discussion on the adaptability of SATCH to task-agnostic or domain-incremental learning settings would broaden its impact.

3. Risk of Overfitting: The assistant teacher’s narrow focus on individual tasks may risk overfitting to specific task features. This might limit the generalization of the student model across a sequence of tasks, particularly if the approach is applied in less controlled or highly variable environments. To strengthen this argument, the authors can add in experiments to measure the generalization capabilities of the student model when SATCH is applied to more complex and variable task sequences. Additionally, consider discussing possible regularization techniques or adjustments to the assistant teacher’s training to mitigate this risk.


4. Gaps in Theoretical Analysis and Interpretability: The paper could benefit from a stronger analysis of interpretability. The assistant teacher introduces additional decision-making layers that could obscure the interpretability of the student model’s predictions. The reliance on visualizations alone may not provide sufficient insights into the assistant teacher’s effect on the knowledge transfer process. Incorporating quantitative metrics for interpretability, such as measuring feature attribution consistency, would add depth to the understanding of SATCH’s impact. A discussion on the trade-offs between interpretability and model complexity introduced by the assistant teacher would also be valuable.

5. Overall presentation clarity: The overall process-flow is hard to follow, it's unclear what process follows what. For example in Fig. 1 the buffer selection for task t is done prior to learning the about the task t in (c). The following figure makes it confusing. The authors can think about reorganizing the description of the methodology to improve clarity. For example, a step-by-step walkthrough of the process, along with a more intuitive depiction in the figures, would be helpful. Explicitly labeling the sequence of operations and ensuring that all components are described in a logical order would enhance comprehension.


Minor typo:
In Line 191-192, ''allows us to maintain'' is repeated.

**Questions:**

1. The proposed SATCH framework is evaluated primarily in class-incremental learning settings where task boundaries are well-defined. However, in many real-world continual learning scenarios, tasks can be overlapping or not strictly disjoint. Could you elaborate on how SATCH handles such situations where task-specific distinctions blur? Specifically, how does the assistant teacher adapt to or mitigate the challenges of overlapping feature distributions, and what impact does this have on the model’s ability to prevent catastrophic forgetting and maintain effective knowledge transfer?

2. The choice of architecture for the assistant teacher is a critical design decision in SATCH, given its role in capturing task-specific knowledge. Could you provide more details on how the architecture of the assistant teacher is selected? How sensitive is the overall performance of the model to this architectural choice, particularly in terms of balancing efficiency and effectiveness? For practitioners aiming to implement SATCH in different environments, what guidelines or heuristics would you recommend for choosing an appropriate assistant teacher architecture?

3. The title emphasizes the goal of mitigating catastrophic forgetting, but the analysis of forgetting prevention appears less explicit in the main text. Could you clarify or point out where the paper quantifies or analyzes the extent of forgetting reduction achieved by SATCH? For example, do you provide a forgetting metric or compare how much past knowledge retention improves relative to baseline methods? An explicit section or metric-based analysis on forgetting would strengthen the paper’s claims.

4. In Equations 1 and 2, the hyperparameter 𝜆 controls the influence of the assistant teacher’s knowledge transfer through Kullback-Leibler divergence. How do you determine the optimal value for 𝜆 in practice? Is there a systematic approach or empirical method that you suggest for tuning this parameter, especially given the diverse nature of continual learning datasets and tasks? Understanding this would aid practitioners in effectively implementing your method in different settings.

---

> ### Author Response · Authors · 2024-11-22
> **Response to Reviewer kU7R (1/3)**
>
> > The proposed SATCH framework is evaluated primarily in class-incremental learning settings where task boundaries are well-defined. However, in many real-world continual learning scenarios, tasks can be overlapping or not strictly disjoint. Could you elaborate on how SATCH handles such situations where task-specific distinctions blur? Specifically, how does the assistant teacher adapt to or mitigate the challenges of overlapping feature distributions, and what impact does this have on the model’s ability to prevent catastrophic forgetting and maintain effective knowledge transfer?
>
> To evaluate SATCH in a continual learning scenario that is similar to the real-world, we applied the **Generalized Class Incremental Learning (GCIL)** setting [P1] to CIFAR100 (GCIL-CIFAR100). This setup introduces three key challenges:
>
> - Tasks may have overlapping classes.
> - The number of classes varies across tasks.
> - Training instances per task are inconsistent.
>
> We evaluated SATCH under GCIL-CIFAR100 with buffer sizes of 1000 and 5000, comparing the accuracy against baseline methods such as DER++, SSIL, CLS-ER, and ESMER. The results are summarized in the table below:
>
> | Memory Size | 1000 | 5000 |
> | --- | --- | --- |
> | JOINT (Upper bound) | 57.21±1.42 | 57.21±1.42 |
> | SGD (Lower bound) | 10.04±0.21 | 10.04±0.21 |
> | ER | 22.41±0.39 | 30.62±0.26 |
> | ER-ACE | 29.89±0.41 | 34.12±0.12 |
> | DER++ | 30.68±0.37 | 41.32±0.42 |
> | + SATCH (ours) | **37.67±0.15** | **44.23±0.11** |
> | SSIL | - | - |
> | + SATCH (ours) | - | - |
> | CLS-ER | 31.46±0.43 | 40.59±0.55 |
> | + SATCH (ours) | **36.12±0.21** | **42.95±0.41** |
> | ESMER | 30.28±0.52 | 35.63±0.52 |
> | + SATCH (ours) | **32.79±0.42** | **37.83±0.58** |
>
> *Results for SSIL is not included as it requires non overlapping classes between tasks.*
>
> SATCH consistently improves accuracy across all baselines and memory sizes, highlighting SATCH’s effectiveness in addressing the challenges posed by overlapping classes and task variability.
>
> We also evaluated SATCH in noisy-CIL settings [P2], where dataset images are assigned random labels. Table 4 shows that SATCH’s ability to filter out noisy labels improves accuracy across all noise levels. Figure 6 demonstrates that SATCH reduces the number of noisy samples stored in the buffer.
>
> > The choice of architecture for the assistant teacher is a critical design decision in SATCH, given its role in capturing task-specific knowledge. Could you provide more details on how the architecture of the assistant teacher is selected? How sensitive is the overall performance of the model to this architectural choice, particularly in terms of balancing efficiency and effectiveness? For practitioners aiming to implement SATCH in different environments, what guidelines or heuristics would you recommend for choosing an appropriate assistant teacher architecture?
>
> We conducted an ablation study on SATCH using three different backbones for the assistant teacher:
>
> - ResNet18 (full model)
> - Reduced ResNet18 (fewer feature layers)
> - A 3-layer convolutional network
>
> Our results in Table 3 highlight that the capacity gap between the assistant teacher and the student impacts the accuracy of SATCH’s components. The Reduced ResNet18 achieved the best balance, offering the highest accuracy and being more efficient than ResNet18. The 3-layer convolutional network, while smaller and more efficient, showed lower performance due to insufficient capacity to distill task-specific knowledge effectively.
>
> Based on our findings, we recommend: First, choosing an assistant teacher with a similar architecture to the student but with reduced feature layers or parameters. This ensures compatibility during distillation while maintaining efficiency. Second, avoiding a large capacity gap [P3] as large differences in backbone size between the assistant teacher and the student can reduce distillation effectiveness. While smaller models may offer higher diversity, they often need more capacity to distill subtle knowledge effectively.
>
> For example, if the backbone we used for our baselines were transformer-based backbone, using a lightweight ViT backbone may be the most effective for the assistant teacher.

---

> ### Author Response · Authors · 2024-11-22
> **Response to Reviewer kU7R (2/3)**
>
> > The title emphasizes the goal of mitigating catastrophic forgetting, but the analysis of forgetting prevention appears less explicit in the main text. Could you clarify or point out where the paper quantifies or analyzes the extent of forgetting reduction achieved by SATCH? For example, do you provide a forgetting metric or compare how much past knowledge retention improves relative to baseline methods? An explicit section or metric-based analysis on forgetting would strengthen the paper’s claims.
>
> We measure forgetting using the Average Forgetting metric [P4] across various datasets (CIFAR100, TinyImageNet, MiniImageNet) and buffer sizes (1000, 5000) for multiple baseline methods. The table below highlights that SATCH consistently reduces forgetting across all baselines.
>
> | Memory Size | CIFAR100 (1000) | CIFAR100 (5000) | TinyImageNet (1000) | TinyImageNet (5000) | MiniImageNet (1000) | MiniImageNet (5000) |
> | --- | --- | --- | --- | --- | --- | --- |
> | ER  | 46.25±2.32 | 28.15±1.48 | 52.94±0.63 | 34.90±0.29 | 46.60±1.67 | 30.89±1.71 |
> | ER-ACE | 25.42±1.44 | 14.31±0.26 | 30.73±0.68 | 21.22±0.81 | 25.67±0.92 | 21.31±1.10 |
> | DER++  | 32.88±0.49 | 16.01±0.82 | 52.71±0.90 | 29.79±0.76 | 48.05±1.03 | 36.54±0.83 |
> | + SATCH (ours) | **22.34±0.23** | **9.73±0.42** | **30.70±0.71** | **15.16±0.49** | **23.77±1.03** | **12.46±0.67** |
> | SSIL | 19.36±0.24 | 15.07±0.15 | 20.16±0.42 | 14.91±0.41 | 11.58±0.86 | 15.21±0.92 |
> | + SATCH (ours) | **17.30±0.86** | **13.57±0.45** | **17.02±0.67** | **10.41±0.29** | **8.18±1.06** | **11.54±0.52** |
> | CLS-ER | 29.31±0.76 | 13.79±0.01 | 46.71±1.10 | 27.65±0.95 | 40.83±1.48 | 33.03±1.54 |
> | + SATCH (ours) | **14.86±0.19** | **10.07±0.76** | **17.26±0.57** | **16.14±0.66** | **10.03±0.22** | **23.80±0.87** |
> | ESMER  | 29.81±1.13 | 13.82±1.66 | 43.10±0.71 | 27.02±0.22 | 37.01±1.48 | 26.69±0.72 |
> | + SATCH (ours) | **18.08±0.97** | **12.10±0.93** | **20.24±1.02** | **14.96±0.48** | **24.85±0.38** | **21.71±1.11** |
>
>
>
> We also measure Average Forgetting under different levels of label noise (10%, 25%, 50%). SATCH consistently reduces forgetting across all baselines despite high levels of noise.
>
> | Label Noise | CIFAR100 (10%) | CIFAR100 (25%) | CIFAR100 (50%) | TinyImageNet (10%) | TinyImageNet (25%) | TinyImageNet (50%) |
> | --- | --- | --- | --- | --- | --- | --- |
> | ER  | 38.57±1.69 | 38.30±1.52 | 31.22±0.91 | 36.98±1.07 | 35.41±1.01 | 25.79±0.48 |
> | ER-ACE | 18.87±1.02 | 19.89±1.08 | 16.15±0.65 | 26.03±0.49 | 24.89±0.54 | 17.60±0.42 |
> | DER++  | 27.16±0.70 | 34.90±1.43 | 33.29±0.52 | 38.20±0.42 | 40.62±0.45 | 33.77±0.56 |
> | + SATCH (ours) | **12.91±0.34** | **9.43±0.62** | **10.03±0.55** | **18.59±0.29** | **21.33±0.22** | **16.36±0.68** |
> | SSIL  | 12.77±0.38 | 13.33±0.29 | 10.98±0.26 | 14.11±0.56 | 15.23±0.44 | 11.79±0.18 |
> | + SATCH (ours) | **9.00±1.10** | **10.39±0.62** | **10.73±0.68** | **9.34±0.31** | **11.69±0.60** | **10.71±0.59** |
> | CLS-ER  | 21.56±0.29 | 26.52±0.74 | 29.30±1.13 | 30.67±0.33 | 34.05±0.25 | 26.80±0.65 |
> | + SATCH (ours) | **8.36±0.40** | **8.69±0.82** | **8.35±0.70** | **13.61±0.44** | **16.87±0.27** | **15.63±0.44** |
> | ESMER  | 15.99±1.34 | 15.66±0.63 | 14.39±0.17 | 24.97±0.59 | 25.69±1.36 | 21.60±0.33 |
> | + SATCH (ours) | **8.48±0.12** | **6.97±0.31** | **6.00±0.55** | **12.34±0.58** | **12.54±0.29** | **10.36±0.14** |

---

> ### Author Response · Authors · 2024-11-22
> **Response to Reviewer kU7R (3/3)**
>
> > In Equations 1 and 2, the hyperparameter 𝜆 controls the influence of the assistant teacher’s knowledge transfer through Kullback-Leibler divergence. How do you determine the optimal value for 𝜆 in practice? Is there a systematic approach or empirical method that you suggest for tuning this parameter, especially given the diverse nature of continual learning datasets and tasks? Understanding this would aid practitioners in effectively implementing your method in different settings.
>
> To provide insight on how to select the optimal λ hyperparameter in practice. We tuned λ using ESMER on CIFAR100 with a buffer size 5000, evaluating four values: [0.1, 0.4, 0.7, 1.0]. The accuracy is presented below:
>
> |  λ | |
> | --- | --- |
> | 1 | 58.28±0.29 |
> | 0.7 | 58.99±0.36 |
> | 0.4 | 59.40±0.25 |
> | 0.1 | **59.97±0.18** |
>
> Our findings show that lower values of λ generally result in higher accuracy, as they reduce the risk of overfitting to the assistant teacher's knowledge. Setting λ too high can lead to overfitting the teacher model's knowledge [2]. In practice, λ can be further tuned per dataset and may achieve a higher accuracy; however, to reduce the dependency on extensive hyperparameter tuning, we set λ = 0.1.
>
> [P1] Mi, F., Kong, L., Lin, T., Yu, K., & Faltings, B. (2020). Generalized class incremental learning. In *Proceedings of the IEEE/CVF conference on computer vision and pattern recognition workshops* (pp. 240-241).
>
> [P2] Sarfraz, F., Arani, E., & Zonooz, B. Error Sensitivity Modulation based Experience Replay: Mitigating Abrupt Representation Drift in Continual Learning. In *The Eleventh International Conference on Learning Representations*.
>
> [P3] Son, W., Na, J., Choi, J., & Hwang, W. (2021). Densely guided knowledge distillation using multiple teacher assistants. In *Proceedings of the IEEE/CVF International Conference on Computer Vision* (pp. 9395-9404).
>
> [P4] Chaudhry, A., Dokania, P. K., Ajanthan, T., & Torr, P. H. (2018). Riemannian walk for incremental learning: Understanding forgetting and intransigence. In *Proceedings of the European conference on computer vision (ECCV)* (pp. 532-547).)

---

> ### Author Response · Authors · 2024-11-26
> **Gentle Reminder**
>
> We sincerely thank you once again for your valuable and constructive feedback. As a gentle reminder, the discussion period will close in approximately one week. We would be happy to further discuss any unresolved questions that you may have.

---

> > ### Comment · Reviewer_kU7R · 2024-11-28
> >
> > I would like to thank the authors' for their response to the initial review. However, I noticed that the concerns raised in the weaknesses section have not been discussed/addressed point-by-point. Could the authors provide detailed responses to the specific weaknesses discussed?

---

> > > ### Author Response · Authors · 2024-11-30
> > > **Response to Weaknesses Reviewer kU7R (1/4)**
> > >
> > > > W1: Limited Discussion on Computational Overheads: The assistant teacher’s additional computations may raise concerns for resource-constrained environments and by makes existing methods computationally inefficient. In additition, the assistant teacher training followed by the distillation performed, makes the knowledge transfer process cumbersome. Adding a detailed analysis of the computational complexity and runtime of SATCH compared to baseline methods. Quantifying the impact on memory and processing time across various settings would clarify the scalability of the approach. Additionally, consider exploring potential optimizations to make the process more efficient, such as parallel training strategies, etc.
> > >
> > > Thank you for your feedback, when compared to a similar multi-teacher method ANCL [P1], which also employs an assistant teacher, SATCH is more memory efficient as shown in Table 5 and achieves higher accuracy as shown in Table 6. The faster runtime is due to using a smaller assistant teacher backbone and specializing on the current task prevents the need to replay past examples.
> > >
> > > **Runtime**:
> > >
> > > - **Assistant Teacher Training**: SATCH requires a single forward and backward pass to train the assistant teacher on the current task. In contrast, ANCL performs additional forward passes through the main teacher to retain past knowledge, increasing runtime. Additionally, ANCL typically employs a larger backbone for the assistant teacher compared to SATCH, further extending training time.
> > > - **Guiding new task learning**: SATCH involves one forward pass for distilling knowledge from the assistant teacher to the student.
> > > - **Diversifying old knowledge:** SATCH avoids additional forward passes during distillation by storing the assistant teacher’s output logits with the buffer samples (as shown in Figure 2(b))
> > >
> > > **Memory**:
> > >
> > > - **Smaller Backbone**: SATCH’s assistant teacher uses a smaller backbone than the student (e.g., a reduced ResNet-18) instead of cloning the student model. SATCH reduces trainable parameters by approximately 11% compared to ANCL, duplicating the backbone of the student.
> > > - **Logit Storage:** Storing the assistant teacher’s logits in the buffer adds a slight memory overhead. For instance, in a dataset with 100 classes and tasks of 10 classes each, only 10 logits are stored per sample, reducing storage compared to retaining all logits.
> > >
> > > **Optimization Strategies**:
> > >
> > > - A smaller backbone for the assistant teacher reduces trainable parameters.
> > > - Task-specific logits are stored with buffer samples, eliminating the need to retain assistant teachers for past tasks.
> > > - The assistant teacher requires less runtime by avoiding knowledge distillation during training.
> > > - Future optimization will focus on reducing replay frequency [P1].
> > >
> > > > W2: The contributions in the paper are limited to a particular kind class-incremental paradigm, therefore its applicability in a broader context remains limited.
> > >
> > > We appreciate the suggestion to expand the settings and paradigms to demonstrate the robustness of SATCH. We have evaluated and performed further evaluation with SATCH on the following settings:
> > >
> > > - **Class incremental learning setting**: Results are presented in Table 1, highlighting SATCH’s accuracy improvements over baselines.
> > > - **Noisy class incremental learning:** A challenging setting incorporating symmetric label noise. Results for this scenario are provided in Table 4, highlighting SATCH’s accuracy improvements over baselines.
> > > - **Generalized class incremental learning:** (Newly added to the paper) A near real-world setting introducing class overlap, varied classes per task, and varied number of training instances. Results are shown in the response to Q1 above.
> > > - **Domain incremental learning**: (Future work) on CLAD-D with self-driving car object classification across day and night domains.

---

> > > ### Author Response · Authors · 2024-11-30
> > > **Response to Weaknesses Reviewer kU7R (2/4)**
> > >
> > > Response to Weaknesses Reviewer kU7R (2/4)
> > >
> > > > W2: To strengthen the argument for SATCH’s effectiveness, The authors could expand the comparative study to include more diverse continual learning approaches, such as parameter isolation techniques (e.g., Progressive Neural Networks or Elastic Weight Consolidation)
> > >
> > > We have expanded our comparisons to include additional regularization methods, such as Function Distance Regularization (FDR) [P4] and Prototype Augmentation and Self-Supervision (PASS) [P5].
> > >
> > > 1. FDR: Saves the network response at the task boundaries and adds a consistency loss on top of ER
> > > 2. PASS: Uses prototypes as feature anchors and incorporates self-supervised auxiliary tasks for feature robustness.
> > >
> > > | Dataset | CIFAR100 | TinyImageNet | MiniImageNet |
> > > | --- | --- | --- | --- |
> > > | FDR  | 41.1±0.52 | 27.22±0.36 | 32.47±0.24 |
> > > | PASS | 48.34±0.92 | 41.18±0.88 | 36.48±1.12 |
> > >
> > > Other methods, such as dynamic architecture techniques, were not included as these methods often grow linearly with the number of tasks [P2], and parameter isolation techniques often require the task-id at inference [P3]. For instance, on CIFAR100, a dynamic architecture method DER [P2] achieves 75.4% accuracy after learning all tasks, but the model parameters increased from 11M to 111M after ten tasks. In contrast, SATCH combined with DER++ achieves 59.97% accuracy using only 12M parameters after learning all tasks. Also, pre-trained models may assume that the labels learned are correct. However, this may not be the case in the noisy class incremental setting [P4]. When using a method proposed by Wu et al. [P5], which freezes network layers to preserve learned features, the model may be unable to recover from the noisy representations learned.
> > >
> > > > W2: Furthermore, a discussion on the adaptability of SATCH to task-agnostic or domain-incremental learning settings would broaden its impact.
> > >
> > > For future work, we plan to adapt SATCH to domain-incremental settings, such as self-driving car object classification in the CLAD-D dataset. Domain-incremental learning involves classifying objects (e.g., cars, pedestrians) under varying domains, such as day and night. SATCH’s ability to store task-specific logits can complement the generalizations of the main teacher, enabling it to retain task-specific complementary features for each domain. This can include changes such as lighting conditions where at night there is reduced illumination of the features, other effects such as glare, and changes in features such as headlights and taillights being more reflective while other features become less prominent.
> > >
> > > > W3: The assistant teacher’s narrow focus on individual tasks may risk overfitting to specific task features. This might limit the generalization of the student model across a sequence of tasks, particularly if the approach is applied in less controlled or highly variable environments
> > >
> > > To clarify, the assistant teacher in SATCH is designed to specialize in task-specific knowledge, complementing the main teacher’s role of generalizing across tasks. Thus, the assistant teacher remembers task-specific features without generalising across multiple tasks.
> > >
> > > > W3: To strengthen this argument, the authors can add in experiments to measure the generalization capabilities of the student model when SATCH is applied to more complex and variable task sequences
> > >
> > > To evaluate the impact of more complex tasks with SATCH, we simulated a larger dataset by doubling the size of each task in TinyImageNet, reducing the total number of tasks from 10 to 5. The table below shows the results of this experiment, comparing the performance of DER++, CLS-ER, and ESMER with and without SATCH for TinyImageNet with 5 tasks:
> > >
> > > | Buffer Size | 1000 | 5000 |
> > > | --- | --- | --- |
> > > | DER++ | 25.88±0.83 | 42.74±0.62 |
> > > | + SATCH (ours) | **40.23±0.26** | **48.72±0.15** |
> > > | CLS-ER | 26.24±0.21 | 42.10±0.71 |
> > > | + SATCH (ours) | **44.43±0.15** | **51.09±0.35** |
> > > | ESMER | 36.10±0.55 | 46.95±0.48 |
> > > | + SATCH (ours) | **45.43±0.62** | **50.03±0.39** |
> > >
> > > The results highlight that SATCH consistently improves accuracy across all baselines and memory sizes. This initially results in SATCH’s ability to effectively leverage task-specific and diverse knowledge even in larger, more complex tasks.

---

> > > ### Author Response · Authors · 2024-11-30
> > > **Response to Weaknesses Reviewer kU7R (4/4)**
> > >
> > > > W5: Overall presentation clarity: The overall process-flow is hard to follow, it's unclear what process follows what. For example in Fig. 1 the buffer selection for task t is done prior to learning the about the task t in (c). The following figure makes it confusing. The authors can think about reorganizing the description of the methodology to improve clarity. For example, a step-by-step walkthrough of the process, along with a more intuitive depiction in the figures, would be helpful. Explicitly labeling the sequence of operations and ensuring that all components are described in a logical order would enhance comprehension.
> > >
> > > We appreciate this feedback and agree that a clearer depiction of the process flow is important for better understanding. We have revised the captions for Fig. 1 to ensure that the logical order of operations is clearly conveyed.
> > >
> > > 1. (a) When learning a new task $t$, an input image $x$…
> > > 2. (b) After generating predictions for input image $x$…
> > > 3. (c) To reduce forgetting of past tasks, buffer samples…
> > >
> > > We will also plan to add a step-by-step walkthrough of the process in the appendix.
> > >
> > > [P1] Smith, J. S., Valkov, L., Halbe, S., Gutta, V., Feris, R., Kira, Z., & Karlinsky, L. (2024). Adaptive Memory Replay for Continual Learning. In Proceedings of the IEEE/CVF Conference on Computer Vision and Pattern Recognition (pp. 3605-3615).
> > >
> > > [P2] Yan, S., Xie, J., & He, X. (2021). Der: Dynamically expandable representation for class incremental learning. In *Proceedings of the IEEE/CVF conference on computer vision and pattern recognition* (pp. 3014-3023).
> > >
> > > [P3] Kang, H., Mina, R. J. L., Madjid, S. R. H., Yoon, J., Hasegawa-Johnson, M., Hwang, S. J., & Yoo, C. D. (2022, June). Forget-free continual learning with winning subnetworks. In *International Conference on Machine Learning* (pp. 10734-10750). PMLR.
> > >
> > > [P4] Sarfraz, F., Arani, E., & Zonooz, B. Error Sensitivity Modulation based Experience Replay: Mitigating Abrupt Representation Drift in Continual Learning. In The Eleventh International Conference on Learning Representations.
> > >
> > > [P5] Wu, T. Y., Swaminathan, G., Li, Z., Ravichandran, A., Vasconcelos, N., Bhotika, R., & Soatto, S. (2022). Class-incremental learning with strong pre-trained models. In *Proceedings of the IEEE/CVF Conference on Computer Vision and Pattern Recognition* (pp. 9601-9610).
> > >
> > > [P6] Sundararajan, M., & Najmi, A. (2020, November). The many Shapley values for model explanation. In *International conference on machine learning* (pp. 9269-9278). PMLR.

---

> > > ### Author Response · Authors · 2024-12-02
> > > **Gentle Reminder**
> > >
> > > We sincerely thank you once again for your valuable and constructive feedback to improve our work. As a gentle reminder, the discussion period will close shortly. We would be happy to further discuss any unresolved questions that you may have.

---

> ### Author Response · Authors · 2024-11-30
> **Response to Weaknesses Reviewer kU7R (3/4)**
>
> >  W4: The assistant teacher introduces additional decision-making layers that could obscure the interpretability of the student model’s predictions.
>
> SATCH is designed as a modular framework that integrates seamlessly with existing knowledge distillation methods. This modularity allows SATCH to be analyzed in isolation or alongside baseline knowledge distillation methods. Additionally, SATCH's components can be applied in any combination, enabling analysis of each component and their interactions. Also, the backbone for the assistant teacher can be varied to understand the effect of different backbone architectures.
>
> > W4: The reliance on visualizations alone may not provide sufficient insights into the assistant teacher’s effect on the knowledge transfer process. Incorporating quantitative metrics for interpretability, such as measuring feature attribution consistency, would add depth to the understanding of SATCH’s impact.
>
> We conducted various experiments to understand further the impact of SATCH on the knowledge distillation process:
>
> - To evaluate SATCH’s effect on different knowledge distillation techniques, we compared the performance of four baseline methods with and without SATCH, as presented in Table 2.
> - To examine the effect of how different assistant teacher backbones affect the accuracy of SATCH’s components in Table 3, we varied the backbone to be the same as the main teacher (ResNet-18) and a smaller backbone (reduced ResNet-18) .
>
> We further investigated the feature attribution consistency using SHAP values [P6]. Cosine similarity was employed to measure the alignment between their feature vectors on CIFAR100, using a randomly selected batch of 32 instances from the first task. The results are summarized below:
>
> |  | SHAP Cosine Similarity  |
> | --- | --- |
> | Student ↔ Main Teacher | 0.51±0.12 |
> | Main Teacher ↔ SATCH Assistant Teacher | 0.07±0.09 |
> - The **student and main teacher's** feature attributions have a cosine similarity of **0.51**, which aligns with expectations. The main teacher's role is to distill knowledge from previous tasks to guide the student's learning and ensure consistency with past task knowledge.
> - The **main teacher and SATCH's assistant teacher** have a cosine similarity of **0.07**, indicating that their feature attributions are minimally correlated. This correlation suggests that the assistant teacher provides different feature representations that complement the main teacher's contributions. This difference likely contributes to the accuracy improvements of SATCH, as the assistant teacher introduces complementary task-specific information that improves the student's learning process.
>
> > W4: A discussion on the trade-offs between interpretability and model complexity introduced by the assistant teacher would also be valuable.
>
> SATCH utilizes a smaller backbone, enhancing interpretability by reducing the number of parameters compared to methods like ANCL. In contrast to ANCL, which employs a larger backbone and retains past-task knowledge, SATCH fine-tunes the assistant teacher exclusively on the current task, simplifying the training process while maintaining task-specific focus.  Additionally, SATCH's components can be applied in any combination, enabling analysis of each component and their interactions independently.

---

### Official Review · Reviewer_yDJ1 · 2024-11-02

**Soundness:** 3
**Presentation:** 3
**Contribution:** 2
**Rating:** 3
**Confidence:** 4

**Summary:**

The paper proposes a more sophisticated knowledge distillation method using an assistant teacher to help transfer knowledge and mitigate catastrophic forgetting in class incremental learning.

**Strengths:**

S1. The proposed method seems to be new and can improve knowledge distillation for class incremental learning.

S2. The writing is generally clear, though, in some places, the paper assumes the reader has prior knowledge of some existing distillation methods.

**Weaknesses:**

W1. The proposed approach is not too novel, as knowledge distillation-based methods are already widely explored, and like this method, do not achieve SOTA performance.

W2. The related work section primarily focuses on distillation-based methods. However, as the proposed approach competes with all existing methods, a more comprehensive review is necessary. The current section may give the impression that the authors are not fully up-to-date with the latest advancements in continual learning.

W3. Paper [a] suggests that catastrophic forgetting may not be the only challenge in class incremental learning. The issue of inter-task class separation is also, maybe more, critical. How can the proposed method deal with that?

W4. The baseline methods are weak and not diverse enough. Other SOTA approaches should also be compared. Please compare with [a, b, c, d]. It appears that the results in [a] are significantly better than those of your proposed method (“ours”), and [a] achieves this without saving any replaying data. The other three systems seem to be strong too.

W5. Nowadays, it’s more appropriate to use a pre-trained model, as it can yield significantly better results. When a pre-trained model is used, knowledge distillation may be less effective because the main feature knowledge is already in the pre-trained model.

[a] A Theoretical Study on Solving Continual Learning. NeurIPS-2022.

[b] DER: Dynamically expandable representation for class incremental learning. CVPR-2021.

[c] BEEF: Bi-compatible class-incremental learning via energy-based expansion and fusion. ICLR 2023.

[d] Prototype augmentation and self-supervision for incremental learning. CVPR-2021

**Questions:**

No questions.

---

> ### Author Response · Authors · 2024-11-22
> **Response to Reviewer yDJ1 (1/2)**
>
> > The proposed approach is not too novel, as knowledge distillation-based methods are already widely explored, and like this method, do not achieve SOTA performance.
>
> Thank you for suggesting a comparison with more SOTA methods. Many of these methods use dynamic architectures or pre-trained models to achieve high accuracy, but often at the cost of increased model complexity, which can grow linearly with the number of tasks learned [P1] that may not be feasible in limited memory constraints. Also, pre-trained models may struggle in scenarios with noisy labels as which may not always be available in real-world applications. We further expand on other SOTA methods and their limitations in response to W4.
>
> While knowledge distillation methods may not achieve SOTA performance in every setting, they remain highly competitive under limited memory constraints. SATCH builds on these methods by improving their effectiveness in challenging, real-world scenarios, such as overlapping classes, noisy environments, and restricted memory.
>
> > The related work section primarily focuses on distillation-based methods. However, as the proposed approach competes with all existing methods, a more comprehensive review is necessary. The current section may give the impression that the authors are not fully up-to-date with the latest advancements in continual learning.
>
> We thank the reviewer for their valuable feedback. We agree that a more comprehensive review of pre-trained and dynamic architecture-based approaches could provide additional context. However, due to space constraints, our related work section prioritizes methods closely aligned with our focus on distillation-based continual learning.
>
> Our work specifically adapts insights from multi-teacher distillation to address real-world continual learning challenges, such as limited memory and noisy environments. This focus was used to prioritize related work in these areas.
>
> > Paper [a] suggests that catastrophic forgetting may not be the only challenge in class incremental learning. The issue of inter-task class separation is also, maybe more, critical. How can the proposed method deal with that?
>
> We thank the reviewer for raising the critical issue of inter-task class separation. Below, we compare the approach of Kim et al. [a], which creates separate networks per task with knowledge distillation methods like SATCH, particularly in real-world settings.
>
> As mentioned by Kim et al. [a], inter-task class separation involves establishing decision boundaries between new and previous task classes. Inter-task class separation is a difficult problem in class incremental learning because past data is unavailable. They decompose the problem into within-task separation and task-id prediction. While this effectively prevents forgetting past classes, it often relies on training a separate model per task, which is then frozen to prevent parameter updates when learning a new task. However, freezing models can limit knowledge transfer as more tasks are learned and may be difficult in real-world scenarios. Also, in real-world scenarios with overlapping classes (where a class may appear in multiple tasks), the task-ID predictor relies on out-of-distribution detection to identify the correct task. However, this can fail when a class exists in multiple task-specific models, leading to incorrect task assignments
>
> Another way to tackle inter-task class separation is knowledge distillation. It inherently deals with this issue by encouraging the current model to mimic the outputs of its previous state, thereby regularizing weights associated with past tasks while learning new ones. This makes distillation methods more robust to real-world scenarios, such as overlapping classes and limited memory availability.
>
> However, distillation techniques lose task-specific knowledge as the model has to generalize to more tasks. SATCH addresses this issue by introducing an assistant teacher trained exclusively on the current task, providing a complementary task-specific perspective. This additional view augments the teacher in guiding the student to improve inter-task class separation without requiring frozen models or explicit task-ID prediction.

---

> ### Author Response · Authors · 2024-11-22
> **Response to Reviewer yDJ1 (2/2)**
>
> > The baseline methods are weak and not diverse enough. Other SOTA approaches should also be compared. Please compare with [a, b, c, d]. It appears that the results in [a] are significantly better than those of your proposed method (“ours”), and [a] achieves this without saving any replaying data. The other three systems seem to be strong too.
>
> Thank you for suggesting a comparison with more SOTA methods such as [a, b ,c, d]. Many of these methods leverage dynamic architectures or pre-trained models to achieve high accuracy, but often at the cost of increased model complexity, which can grow linearly with the number of tasks learned [a, b, c].
>
> WPTP [a] achieves SOTA accuracy by creating a subnetwork per task and determines which subnetwork classifies a new image by predicting the task ID at inference using out-of-distribution techniques. However, prediction time scales linearly with the number of tasks, as each task-specific model must determine if the input is out of distribution. WPTP requires non-overlapping classes and clean data, which limits its applicability to real-world settings like those with overlapping classes or noisy labels.
>
> DER [b] is a dynamically expanding method that adds new parameters for each task and prevents updates to previously learned weights to avoid forgetting. DER achieves a SOTA accuracy of 75.4% on CIFAR100. However, the parameters grow linearly with the number of tasks, increasing from 11M to 111M after ten tasks. Also, DER assumes no noise exists in the dataset as the subnetworks for previous tasks are frozen after learning. This is the same for the BEEF method [c], as it also trains a new module per task where the model parameters grow linearly.
>
> PASS [d] uses prototypes, which act as anchors for each class in feature space. PASS doesn’t rely on pre-trained models and the model does not grow with the number of tasks. We incorporate PASS in our baselines in Table 1 with the following results:
>
> |  | CIFAR100 | TinyImageNet | MiniImageNet |
> | --- | --- | --- | --- |
> | PASS | 48.34±0.92 | 41.18±0.88 | 36.48±1.12 |
> | ESMER+SATCH | **58.48±0.32** | **47.07±0.28** | **38.84±0.94** |
>
> PASS is a competitive method but has lower accuracy than ESMER + SATCH with 5000 buffer size.
>
> > Nowadays, it’s more appropriate to use a pre-trained model, as it can yield significantly better results. When a pre-trained model is used, knowledge distillation may be less effective because the main feature knowledge is already in the pre-trained model.
>
> Knowledge distillation is a widely used technique in continual learning to reduce forgetting. It introduces a regularization term that encourages the "student" model to mimic the outputs of its previous state, the "teacher," typically created by cloning the student before learning a new task. This process helps retain task-specific knowledge while accommodating new information.
>
> We agree that pre-trained models can significantly improve accuracy due to their ability to extract representative features. For instance, methods like [P2] pre-train models on a large dataset, freezing early feature layers while leaving later layers unfrozen to learn unseen classes. We plan to explore pre-trained models in future work to evaluate SATCH’s performance in such settings.
>
> Pre-trained models can complement knowledge distillation by focusing on general feature initialization, allowing the teacher model to specialize in capturing nuanced, task-specific knowledge. This approach ensures that task-specific insights enhance the generalized knowledge extracted by the pre-trained model. SATCH further improves this process by leveraging its smaller assistant teacher, which provides an additional task-specific perspective.
>
> [P1] Yan, S., Xie, J., & He, X. (2021). Der: Dynamically expandable representation for class incremental learning. In *Proceedings of the IEEE/CVF conference on computer vision and pattern recognition* (pp. 3014-3023).
>
> [P2] Wu, T. Y., Swaminathan, G., Li, Z., Ravichandran, A., Vasconcelos, N., Bhotika, R., & Soatto, S. (2022). Class-incremental learning with strong pre-trained models. In Proceedings of the IEEE/CVF Conference on Computer Vision and Pattern Recognition (pp. 9601-9610).

---

> ### Author Response · Authors · 2024-11-26
> **Gentle Reminder**
>
> We sincerely thank you once again for your valuable and constructive feedback. As a gentle reminder, the discussion period will close in approximately one week. We would be happy to further discuss any unresolved questions that you may have.

---

> > ### Comment · Reviewer_yDJ1 · 2024-12-02
> > **Feedback**
> >
> > Thank you for your detailed responses. Unfortunately, the answers are not entirely satisfactory. For example, distillation does not effectively address inter-task class separation. Regarding overlapping classes, numerous online continual learning methods tackle scenarios with blurry task boundaries, which could be explored in a separate study. Your approach leverages rehearsal data, yet the accuracy results are significantly lower than those achieved by state-of-the-art methods that neither rely on rehearsal data nor pre-trained models. A fundamental challenge in continual learning is its accuracy gap compared to the JOINT method, which significantly hinders its practical applicability. Without achieving competitive accuracy, the potential for meaningful progress remains limited.

---

> > > ### Author Response · Authors · 2024-12-03
> > > **References**
> > >
> > > [P1] Yu, J., Zhuge, Y., Zhang, L., Hu, P., Wang, D., Lu, H., & He, Y. (2024). Boosting continual learning of vision-language models via mixture-of-experts adapters. In *Proceedings of the IEEE/CVF Conference on Computer Vision and Pattern Recognition* (pp. 23219-23230).
> > >
> > > [P2] Kim, G., Xiao, C., Konishi, T., Ke, Z., & Liu, B. (2022). A theoretical study on solving continual learning. *Advances in neural information processing systems*, *35*, 5065-5079.
> > >
> > > [P3] Yan, S., Xie, J., & He, X. (2021). Der: Dynamically expandable representation for class incremental learning. In *Proceedings of the IEEE/CVF conference on computer vision and pattern recognition* (pp. 3014-3023).
> > >
> > > [P4] Wang, F. Y., Zhou, D. W., Liu, L., Ye, H. J., Bian, Y., Zhan, D. C., & Zhao, P. BEEF: Bi-Compatible Class-Incremental Learning via Energy-Based Expansion and Fusion. In *The Eleventh International Conference on Learning Representations*.
> > >
> > > [P5] Huang, L., Cao, X., Lu, H., & Liu, X. (2024, September). Class-Incremental Learning with CLIP: Adaptive Representation Adjustment and Parameter Fusion. In *European Conference on Computer Vision* (pp. 214-231).

---

> ### Author Response · Authors · 2024-12-02
> **Gentle Reminder**
>
> We sincerely thank you for your invaluable suggestions and feedback. As the discussion period is nearing its end, we kindly ask if our previous responses have addressed your concerns.
>
> If you have any additional concerns or questions, please let us know before the rebuttal period ends, and we will be happy to address them.

---

> ### Author Response · Authors · 2024-12-03
> **SATCH State-of-the-Art Transformer Results**
>
> > A fundamental challenge in continual learning is its accuracy gap compared to the JOINT method, which significantly hinders its practical applicability. Without achieving competitive accuracy, the potential for meaningful progress remains limited.
>
> We agree that the accuracy gap compared to the JOINT method is critical for practical applications; that is why we test the accuracy improvements of SATCH when applied to state-of-the-art transformer methods (as requested by reviewer 7tBY). We apply SATCH to Mixture-of-Experts Adapters (MoE) [P1], which use LoRA to learn tasks incrementally while reducing runtime and memory overhead. While MoE reduces computational complexity compared to parameter-isolation methods, it suffers from inter-task class separation due to an expert having to learn multiple tasks incrementally.
>
> We improve inter-task class separation by applying SATCH by distilling past task knowledge as new tasks are learned, improving inter-task class separation and overall accuracy. To evaluate state-of-the-art accuracy, we compare MoE + SATCH against the baselines mentioned in W4 on CIFAR100 using a buffer size of 5000 and λ=0.1. Baseline results are sourced from their respective original papers, and the final accuracy is reported below:
>
> | Methods | Final Accuracy (%) |
> | --- | --- |
> | **MoE + SATCH (Ours)** | **79.78** |
> | MoE (Yu et al. CVPR 24) [P1] | 78.42 |
> | RAPF (Huang et al. ECCV 24) [P5] | 79.04 |
> | BEEF (Wang et al. ICLR 23) [P4] | 72.93 |
> | DER (Yan et al. CVPR 21) [P3] | 69.94 |
> | Sup+CSI (Kim et al. Neurips 22) [P2] | 65.20 |
> |  |  |
>
> SATCH improves MoE accuracy by 1.36% and outperforms RAPF by 0.74%, achieving state-of-the-art performance. This shows that SATCH can effectively improve accuracy of MoE that uses a pre-train transformer based architecture.
>
> To investigate forgetting in MoE adapters, we track accuracy changes over time for the first and second tasks:
>
> | Tasks Learned | 0 | 1 | 2 | 3 | 4 | 5 | 6 | 7 | 8 | 9 |
> | --- | --- | --- | --- | --- | --- | --- | --- | --- | --- | --- |
> | MoE Task 1 Accuracy | **98.5** | 95.3 | 91.0 | 85.5 | 83.0 | 82.3 | 81.3 | 77.4 | 80.1 | 80.5 |
> | + SATCH Task 1 Accuracy | 98.3 | **97.3** | **95.5** | **91.4** | **88.8** | **87.7** | **88.1** | **87.5** | **85.1** | **84.7** |
> | MoE Task 2 Accuracy | **93.2** | 90.3 | 88.1 | 87.4 | 85.5 | 82.5 | 78.9 | 79.6 | 76.9 |  |
> | + SATCH Task 2 Accuracy | 92.6 | **92.6** | **91.0** | **90.5** | **86.0** | **85.9** | **84.2** | **83.7** | **81.0** |  |
>
> These results demonstrate that SATCH can reduce forgetting compared to MoE alone, retaining higher accuracy for earlier tasks as more tasks are learned. This improvement in inter-task class separation addresses a key limitation of MoE.
>
> > For example, distillation does not effectively address inter-task class separation. Your approach leverages rehearsal data, yet the accuracy results are significantly lower than those achieved by state-of-the-art methods that neither rely on rehearsal data nor pre-trained models
>
> The purpose of distillation in continual learning is to mitigate forgetting as new tasks are learned. Parameter-isolation methods, which train and freeze separate networks for each task, inherently avoid forgetting since the weights of previous tasks remain unchanged. However, adapting these methods to class-incremental learning introduces a trade-off, as they require task-ID identification techniques, such as out-of-distribution (OOD) detection, which are computationally intensive.
>
> The purpose of distillation in continual learning is to mitigate forgetting as new tasks are learned. In contrast, parameter-isolation methods, which train and freeze separate networks for each task, inherently avoid forgetting since the weights of previous tasks remain unchanged. However, adapting these methods to class-incremental learning introduces a trade-off, as they require task-ID identification techniques, such as out-of-distribution detection, which is computationally intensive. For example, the method in [a] requires four rotations of a single image, generating predictions for each rotation and combining them to select the most probable task. This process incurs a computational cost of: `test batch size × learned tasks × rotations` forward passes, which scales with the number of tasks and may not be practical with a large number of tasks.
>
> Recent methods aim to reduce computational costs by fusing task-specific models [P5] and training an expert to learn multiple tasks [P1]. However, when an expert model’s parameters change, especially with limited access to past data, forgetting occurs, as shown in our experiments. SATCH is applied to continual learning methods where forgetting occurs and when applied to state-of-the-art transformer models, SATCH helps reduce forgetting to improve state-of-the-art accuracy.

---

### Official Review · Reviewer_dGZY · 2024-11-04

**Soundness:** 2
**Presentation:** 2
**Contribution:** 2
**Rating:** 6
**Confidence:** 3

**Summary:**

The paper presents SATCH (Specialized Assistant Teacher Distillation), a novel approach designed to address catastrophic forgetting in continual learning through a specialized assistant-teacher mechanism. This assistant teacher is trained on individual tasks before the student learns them, providing diverse, task-specific guidance that enhances memory retention and reduces the forgetting of previously learned tasks. Key contributions include (1) guiding new task learning with task-specific soft labels, (2) refining buffer selection to prioritize representative samples, and (3) diversifying knowledge distillation by combining the assistant teacher's specialized knowledge with the main teacher’s generalized knowledge. Experiments on benchmarks like CIFAR-100, TinyImageNet, and MiniImageNet demonstrate significant improvements in continual learning accuracy, particularly in settings with noisy data.

**Strengths:**

1. By introducing a specialized assistant teacher that learns each task individually, SATCH diversifies and enhances knowledge distillation, addressing a significant limitation in existing continual learning frameworks.

2. The buffer selection refinement effectively filters noisy samples, enhancing stability and making the method robust to real-world scenarios with label noise.

3. The paper provides thorough experimental validation across multiple datasets, benchmarking SATCH against established methods. It demonstrates consistent accuracy improvements and provides evidence for reduced catastrophic forgetting.

4. SATCH is designed to integrate seamlessly with various continual learning methods, enhancing its practicality and potential adoption.

**Weaknesses:**

1. While SATCH improves accuracy, it introduces additional computation through the assistant teacher and buffer operations. The paper would benefit from a clearer comparison of the memory and runtime efficiency with alternative methods, especially on larger-scale tasks or models.

2. The assistant teacher’s architecture is described as a scaled-down ResNet-18, which may not generalize well across diverse models or tasks. An analysis of SATCH’s scalability with more complex backbones or larger task sequences would add value.

3. Although the assistant teacher provides task-specific knowledge, the long-term retention of this information across tasks remains under-explored. It would be helpful to see additional studies or visualizations that clarify the assistant teacher’s impact on task-specific feature preservation over extended sequences.

4. The ablation study does not fully explain the contributions of each component in isolation, especially under noisy conditions. More detailed component-wise evaluations would make it easier to understand the relative impact of each part (e.g., buffer selection refinement, diverse knowledge).

**Questions:**

1. How does the assistant teacher impact memory and runtime compared to single-teacher methods on larger datasets?

2. Please consider comparing SATCH with multi-teacher approaches that focus on task-specific retention.

3. It is suggested to evaluate if SATCH handles larger, real-world datasets beyond CIFAR100 and MiniImageNet.

4. It is recommended to perform experiments if SATCH manages cases with overlapping tasks or undefined task boundaries.

5. Is it possible to expand ablation studies to show SATCH’s component performance under varying noise levels and buffer sizes?

6. It is better to add more analysis on how SATCH preserves task-specific knowledge?

7. How sensitive is SATCH to settings like distillation weight and buffer size?

---

> ### Author Response · Authors · 2024-11-22
> **Response to Reviewer dGZY (1/3)**
>
> > How does the assistant teacher impact memory and runtime compared to single-teacher methods on larger datasets?
>
> Our work demonstrates the benefits of multi-teacher-based approaches in continual learning. When compared to a similar multi-teacher method ANCL [P1], which also employs an assistant teacher, SATCH is more memory-efficient due to its smaller assistant teacher backbone and achieves faster runtime since the assistant teacher focuses exclusively on the current task without needing additional components to retain past knowledge.
>
> **Runtime**:
>
> - **Assistant Teacher Training**: SATCH requires a single forward and backward pass to train the assistant teacher on the current task. In contrast, ANCL performs additional forward passes through the main teacher to retain past knowledge, increasing runtime. Additionally, ANCL typically employs a larger backbone for the assistant teacher compared to SATCH, further extending training time.
> - **Guiding new task learning**: SATCH involves one forward pass for distilling knowledge from the assistant teacher to the student.
> - **Diversifying old knowledge:** SATCH avoids additional forward passes during distillation by storing the assistant teacher’s output logits with the buffer samples (as shown in Figure 2(b))
>
> **Memory**:
>
> - **Smaller Backbone**: SATCH’s assistant teacher uses a smaller backbone than the student (e.g., a reduced ResNet-18) instead of cloning the student model. SATCH reduces trainable parameters by approximately 11% compared to ANCL, duplicating the backbone of the student.
> - **Logit Storage:** Storing the assistant teacher’s logits in the buffer adds a slight memory overhead. For instance, in a dataset with 100 classes and tasks of 10 classes each, only 10 logits are stored per sample, reducing storage compared to retaining all logits.
>
> Table 5 quantifies the runtime and memory in terms of memory usage (MB) and runtime (epochs/hour); the experiment compares ANCL with SATCH and highlights the additional runtime and memory required compared to a single teacher.
>
> > Please consider comparing SATCH with multi-teacher approaches that focus on task-specific retention.
>
> Table 6 compares SATCH with ANCL [P1], a multi-teacher approach that employs an additional teacher to guide student learning. Unlike ANCL, which primarily focuses on guiding new task learning, SATCH emphasizes distilling task-specific knowledge lost by the teacher over time and using a smaller backbone to capture a different understanding of the task.
>
> The improvements observed with SATCH over ANCL may be due to the diverse backbone architectures and initialization of the assistant teacher. SATCH uses a smaller backbone for the assistant teacher, whereas ANCL creates the assistant teacher by cloning the student model. Another potential reason why SATCH performs better than ANCL is that the initialization and weights of the assistant teacher are cloned from the student, which may limit the impact of multi-teacher distillation [P2].
>
> > It is suggested to evaluate if SATCH handles larger, real-world datasets beyond CIFAR100 and MiniImageNet.
>
> Thank you for the suggestion to evaluate SATCH on larger, real-world datasets. Given the limited time, we simulated a larger dataset by doubling the size of each task in TinyImageNet, reducing the total number of tasks from 10 to 5.
>
> The table below summarizes the results for this experiment, comparing the performance of DER++, CLS-ER, and ESMER with and without SATCH on TinyImageNet with 5 tasks:
>
> | **Buffer Size** | **1000** | **5000** |
> | --- | --- | --- |
> | DER++ | 25.88±0.83 | 42.74±0.62 |
> | + SATCH (ours) | **40.23±0.26** | **48.72±0.15** |
> | CLS-ER | 26.24±0.21 | 42.10±0.71 |
> | + SATCH (ours) | **44.43±0.15** | **51.09±0.35** |
> | ESMER | 36.10±0.55 | 46.95±0.48 |
> | + SATCH (ours) | **45.43±0.62** | **50.03±0.39** |
>
> The results highlights that SATCH consistently improves accuracy across all baselines and memory sizes. This provides initial results in SATCH’s ability to effectively leverage task-specific and diverse knowledge even in larger and more complex tasks.
>
> In future work, we plan to extend our experiments to additional real-world datasets, such as CLAD-D [P3].

---

> ### Author Response · Authors · 2024-11-22
> **Response to Reviewer dGZY (2/3)**
>
> > It is recommended to perform experiments if SATCH manages cases with overlapping tasks or undefined task boundaries.
>
> To test the robustness, we applied the Generalized Class Incremental Learning setting [P4] to CIFAR100 (GCIL-CIFAR100),  replicating real-world settings. This setting introduces three key challenges:
>
> - Tasks may have overlapping classes.
> - The number of classes varies across tasks.
> - Training instances per task are inconsistent.
>
> We evaluated SATCH under GCIL-CIFAR100 with buffer sizes of 1000 and 5000, comparing the accuracy against baseline methods such as DER++, SSIL, CLS-ER, and ESMER. The results are summarized in the table below:
>
> | **Memory Size** | **1000** | **5000** |
> | --- | --- | --- |
> | JOINT (Upper bound) | 57.21±1.42 | 57.21±1.42 |
> | SGD (Lower bound) | 10.04±0.21 | 10.04±0.21 |
> | ER | 22.41±0.39 | 30.62±0.26 |
> | ER-ACE  | 29.89±0.41 | 34.12±0.12 |
> | DER++ | 30.68±0.37 | 41.32±0.42 |
> | + SATCH (ours) | **37.67±0.15** | **44.23±0.11** |
> | SSIL | - | - |
> | + SATCH (ours) | - | - |
> | CLS-ER | 31.46±0.43 | 40.59±0.55 |
> | + SATCH (ours) | **36.12±0.21** | **42.95±0.41** |
> | ESMER | 30.28±0.52 | 35.63±0.52 |
> | + SATCH (ours) | **32.79±0.42** | **37.83±0.58** |
>
> * Results for SSIL is not included as it can not work with overlapping classes.
>
> SATCH consistently improves accuracy across all baselines and memory sizes, highlighting SATCH’s effectiveness in addressing the challenges posed by overlapping classes and task variability.
>
> > Is it possible to expand ablation studies to show SATCH’s component performance under varying noise levels and buffer sizes?
>
> Below, we present additional results evaluating SATCH’s components under varying noise levels and buffer sizes.
>
> We evaluated SATCH with ESMER on CIFAR100 under 25% and 50% label noise with a buffer size of 5000:
>
> | BIAM | NEWL | DIVK | BUFS | 25% | 50% |
> | --- | --- | --- | --- | --- | --- |
> | ✔ | ✔ | ✔ | ✔ | 44.62±0.39 | 28.53±0.46 |
> | ✔ | ✔ | ✔ | ✖ | 43.79±0.60 | 26.14±0.16 |
> | ✔ | ✖ | ✔ | ✖ | 42.20±0.13 | 24.67±0.37 |
> | ✔ | ✔ | ✖ | ✖ | 43.26±0.63 | 25.53±0.20 |
> | ✔ | ✖ | ✖ | ✖ | 41.71±0.53 | 23.98±0.11 |
> | ✖ | ✖ | ✖ | ✖ | 37.01±0.52 | 20.82±0.33 |
>
> Under 25% noise levels, the buffer selection component (BUFS) has a 0.83% increase compared to a noise level of 2.39% under 50%. This highlights SATCH’s ability to handle noisy environments and select less noisy samples stored in the buffer, as shown in Figure 6.
>
> We also evaluate SATCH with ESMER on CIFAR100 for buffer sizes of 1000 and 5000:
>
> | BIAM | NEWL | DIVK | BUFS | 1000 | 5000 |
> | --- | --- | --- | --- | --- | --- |
> | ✔ | ✔ | ✔ | ✔ | 52.09±0.45 | 58.48±0.32 |
> | ✔ | ✔ | ✔ | ✖ | 51.72±0.15 | 58.22±0.09 |
> | ✔ | ✖ | ✔ | ✖ | 49.95±0.50 | 57.43±0.65 |
> | ✔ | ✔ | ✖ | ✖ | 49.86±0.62 | 57.22±0.30 |
> | ✔ | ✖ | ✖ | ✖ | 47.92±0.39 | 56.11±0.37 |
> | ✖ | ✖ | ✖ | ✖ | 45.55±0.65 | 55.29±0.59 |
>
> SATCH has a larger accuracy improvement on buffer size 1000. For example, BIAM + DIVK show a 4.4% accuracy improvement for a 1000 buffer size versus 2.14% for a 5000 buffer size. These results suggest that the additional knowledge from SATCH is more effective when the buffer size is smaller. In larger buffers, the greater quantity of stored samples may provide sufficient task knowledge, reducing the impact of SATCH.
>
>
>
> > It is better to add more analysis on how SATCH preserves task-specific knowledge?
>
> To clarify, SATCH preserves task-specific knowledge by exclusively training a smaller assistant teacher model on the current task. SATCH focuses only on a single task at a time to learn task-specific features that are then represented by the output logits. These logits are stored in the replay buffer along with the corresponding data instances. The logits remain stable because they are not updated after being stored in the buffer, avoiding forgetting.
>
> In Figure 5(a), we present the task-specific accuracy of SATCH’s assistant teacher compared to baseline methods. SATCH maintains task-specific accuracy across tasks, while baseline methods show a decline in task-specific accuracy as the main teacher model has to generalize across tasks.

---

> ### Author Response · Authors · 2024-11-22
> **Response to Reviewer dGZY (3/3)**
>
> > How sensitive is SATCH to settings like distillation weight and buffer size?
>
> The λ hyperparameter in Equation 2 is a weight that controls the assistant teacher's influence during new task learning. We tuned λ using ESMER on CIFAR100 with a buffer size of 5000, evaluating four values: [0.1, 0.4, 0.7, 1.0]. The accuracy is presented below:
>
> | λ |  |
> | --- | --- |
> | 1 | 58.28±0.29 |
> | 0.7 | 58.99±0.36 |
> | 0.4 | 59.40±0.25 |
> | 0.1 | **59.97±0.18** |
>
> Our findings show that lower values of λ generally result in higher accuracy, as they reduce the risk of overfitting to the assistant teacher’s knowledge. Setting λ too high can lead to overfitting on the teacher model's knowledge [P5]. Based on this insight, we set λ = 0.1 for all buffer sizes and datasets to reduce the method’s dependency on extensive hyperparameter tuning.
>
> SATCH’s sensitivity to buffer size can be seen in the ablation study in question 5. SATCH has a higher accuracy improvement with the smaller buffer size of 1000 compared to the larger size of 5000.
>
> [P1] Kim, S., Noci, L., Orvieto, A., & Hofmann, T. (2023). Achieving a better stability-plasticity trade-off via auxiliary networks in continual learning. In Proceedings of the IEEE/CVF Conference on Computer Vision and Pattern Recognition (pp. 11930-11939).
>
> [P2] Wang, L., & Yoon, K. J. (2021). Knowledge distillation and student-teacher learning for visual intelligence: A review and new outlooks. IEEE transactions on pattern analysis and machine intelligence, 44(6), 3048-3068.
>
> [P3] Verwimp, E., Yang, K., Parisot, S., Hong, L., McDonagh, S., Pérez-Pellitero, E., ... & Tuytelaars, T. (2023). Clad: A realistic continual learning benchmark for autonomous driving. Neural Networks, 161, 659-669.
>
> [P4] Mi, F., Kong, L., Lin, T., Yu, K., & Faltings, B. (2020). Generalized class incremental learning. In *Proceedings of the IEEE/CVF conference on computer vision and pattern recognition workshops* (pp. 240-241).
>
> [P5] Hinton, G., Vinyals, O., & Dean, J. (2015). Distilling the Knowledge in a Neural Network. *stat*, *1050*, 9.

---

> ### Author Response · Authors · 2024-11-26
> **Gentle Reminder**
>
> We sincerely thank you once again for your valuable and constructive feedback. As a gentle reminder, the discussion period will close in approximately one week. We would be happy to further discuss any unresolved questions that you may have.

---

> ### Comment · Reviewer_dGZY · 2024-12-02
> **Final Rating from Reviewer dGZY**
>
> I appreciate the efforts and responses made by the authors. I would like to raise my score to marginally above the acceptance threshold.

---

> > ### Author Response · Authors · 2024-12-02
> > **Official Comment by Authors**
> >
> > Thank you for the constructive discussion and insights that have allowed us to improve the quality of our paper. If you have any additional concerns or questions, please let us know before the rebuttal period ends, and we will be happy to address them.

---

### Official Review · Reviewer_7tBY · 2024-11-04

**Soundness:** 3
**Presentation:** 3
**Contribution:** 2
**Rating:** 6
**Confidence:** 3

**Summary:**

The paper introduces a new class incremental continual learning framework. It uses an assistant teacher network to diversify knowledge but also be task-specific. The logits from the assistant teacher is stored in the memory buffer which is used to diversify the knowledge during the knowledge distillation process. It also uses a buffer selection strategy to keep representative samples in the memory buffer. The experiments show that these steps improve the accuracy and reduce catastrophic forgetting.

**Strengths:**

- The proposed idea sounds and improves multiple baseline models.
- The paper is clearly written, especially Figure 2 is very informative.
- Grad-CAM visualization and the ablation study show the benefit of the proposed method.

**Weaknesses:**

1. Since the proposed method has an additional model (assistant teacher), this adds additional parameters to the framework. Ideally, the total model size should match all models for a fair comparison. What is the total model size for all models? I suggest the authors report the current parameter counts and provide the comparison with equal total model size for all methods if possible—enlarge the models to match total parameter counts.

2. The paper claims that combining the logits of the replay buffer and the teacher diversifies the knowledge. It is unclear how this step helps diversify knowledge. Could authors explain this? Also, I ask the authors to provide quantitative metrics or visualizations that demonstrate increased diversity in the combined knowledge compared to using only the main teacher or replay buffer logits.

3. To understand how good the proposed method is, I suggest authors provide the upper and lower bounds --- training all tasks jointly (upper bound) or sequentially (lower bound) without any techniques.

4. Figures 3 and 5 results are with a buffer size of 1000, and Tables 2-4 are with a buffer size of 5000. Could the authors either provide results for both buffer sizes consistently across all experiments or explain their rationale for using different buffer sizes in different analyses?

5. The choice of backbone: The backbones the authors tested are ResNet-18 and 3-layer convnet. Are there any potential challenges or modifications needed to apply SATCH to transformer-based architectures? Could the authors provide preliminary results with a transformer-based architecture like ViT if feasible?

6. Also, it would be better to compare with more recent SOTA and other class incremental learning methods such as [1-3].

[1] Class-Incremental Learning With Strong Pre-Trained Models, CVPR 2022
[2] DyTox: Transformers for Continual Learning With DYnamic TOken eXpansion, CVPR 2022
[3] Class-incremental learning with clip: Adaptive representation adjustment and parameter fusion, ECCV 2024

**Questions:**

See the weaknesses.

---

> ### Author Response · Authors · 2024-11-22
> **Response to Reviewer 7tBY (1/3)**
>
> > Since the proposed method has an additional model (assistant teacher), this adds additional parameters to the framework. Ideally, the total model size should match all models for a fair comparison. What is the total model size for all models? I suggest the authors report the current parameter counts and provide the comparison with equal total model size for all methods if possible—enlarge the models to match total parameter counts.
>
> We thank the reviewer for highlighting the importance of parameter fairness across models. To address this concern, we calculated the total trainable parameters for the backbones used in our experiments and adjusted the `nf` hyperparameter, which controls the initial number of convolutional filters, to align model sizes.
>
> The table below summarizes the trainable parameter counts:
>
> | **Backbone** | **Trainable Parameters** |
> | --- | --- |
> | ResNet18 (nf=64) | 11.40M |
> | Reduced ResNet (nf=20) | 1.25M |
> | ResNet18 + Reduced ResNet | 12.65M |
> | Enlarged ResNet18 (nf=68) | 12.85M |
>
> In our experiments, we used ResNet18 (nf=64) as the backbone for the reported baselines (11.40M parameters) and Reduced ResNet (1.25M parameters, 11% of the baseline size) for SATCH’s assistant teacher. Combining SATCH with a baseline requires a total of 12.65M parameters. To ensure a fair comparison, we enlarged ResNet18 further by increasing the convolutional filters (nf=68) to match the parameter count of our approach. This configuration is referred to as **Enlarged ResNet18**.
>
> We evaluated the accuracy of our SATCH method compared to the standard and enlarged baseline models on CIFAR100. The results, averaged over three runs, are presented below:
>
> | **Memory Size** | **1000** | **5000** |
> | --- | --- | --- |
> | DER++_enlarged | 45.49±0.63 | 57.80±0.73 |
> | DER++ | 44.62±0.56 | 56.39±1.06 |
> | + SATCH (ours) | **48.38±0.19** | **59.97±0.18** |
> | SSIL_enlarged | 41.17±0.49 | 52.61±1.01 |
> | SSIL | 40.70±0.40 | 51.54±0.89 |
> | + SATCH (ours) | **42.95±0.17** | **54.06±0.64** |
> | CLS-ER_enlarged | 47.44±0.59 | 60.02±0.95 |
> | CLS-ER | 45.47±0.63 | 59.63±1.12 |
> | + SATCH (ours) | **52.36±0.30** | **61.39±0.30** |
> | ESMER_enlarged | 45.95±0.49 | 56.82±0.64 |
> | ESMER | 45.55±0.65 | 55.29±0.59 |
> | + SATCH (ours) | **52.09±0.68** | **58.48±0.32** |
>
> The findings demonstrate that while enlarging the ResNet18 backbone improves accuracy compared to the standard ResNet18 backbone, SATCH consistently outperforms all baselines.
>
> > The paper claims that combining the logits of the replay buffer and the teacher diversifies the knowledge. It is unclear how this step helps diversify knowledge. Could authors explain this?
>
> We clarify the definition of diverse knowledge and how combining logits from the replay buffer and the main teacher diversifies the knowledge.
>
> Diverse knowledge in our study refers to broader and more generalized feature representations that enable the student model to retain prior knowledge while reducing overfitting to specific tasks. Previous work in multi-teacher distillation demonstrates that students can achieve higher accuracy when learning from multiple teachers as they contribute a unique perspective that is more informative than a single teacher [P1, P6]. SATCH is inspired by this insight by introducing a specialized assistant teacher with a smaller backbone specializing in a single task suited for the continual learning setting. This assistant teacher focuses on task-specific details, complementing the generalized representations of the main teacher.
>
> In SATCH, the logits stored in the replay buffer capture task-specific information generated by the assistant teacher, which serves as a proxy for diverse knowledge. When combined with the main teacher's generalized logits, they produce a representation that balances task-specific and generalized information. This approach enables the student model to benefit from a more comprehensive understanding of specific and generalized patterns.
>
> > Also, I ask the authors to provide quantitative metrics or visualizations that demonstrate increased diversity in the combined knowledge compared to using only the main teacher or replay buffer logits.
>
> Figure 1 (Grad-CAM) illustrates the diversity in knowledge transfer. For example, on a polar bear input, the assistant teacher highlights task-specific features, such as the ears and legs, while the main teacher focuses on more generalized features, such as the eyes. When these logits are combined, the resulting representation exhibits a richer feature map with broader activations, combining complementary insights from both teachers.

---

> ### Author Response · Authors · 2024-11-22
> **Response to Reviewer 7tBY (2/3)**
>
> > To understand how good the proposed method is, I suggest authors provide the upper and lower bounds --- training all tasks jointly (upper bound) or sequentially (lower bound) without any techniques.
>
> Below, we report the results for all tasks trained jointly (JOINT, upper bound) and sequentially without any techniques (SGD, lower bound) on CIFAR100, TinyImageNet, and MiniImageNet:
>
> |  | CIFAR100 | TinyImageNet | MiniImageNet |
> | --- | --- | --- | --- |
> | JOINT | 70.11±0.21 | 59.69±0.13 | 45.40±0.09 |
> | SGD | 9.34±0.05 | 8.12±0.08 | 9.28±0.06 |
>
> We have also added JOINT and SGD into our experiments for the noisy class incremental setting:
>
> | Label Noise | CIFAR100 (10%) | CIFAR100 (25%) | CIFAR100 (50%) | TinyImageNet (10%) | TinyImageNet (25%) | TinyImageNet (50%) |
> | --- | --- | --- | --- | --- | --- | --- |
> | JOINT | 62.86±0.41 | 59.09±0.27 | 51.62±0.39 | 50.62±0.19 | 45.82±0.51 | 40.09±0.26 |
> | SGD | 7.44±0.38 | 6.52±0.40 | 5.31±0.33 | 6.66±0.18 | 5.51±0.11 | 3.42±0.04 |
>
> > Figures 3 and 5 results are with a buffer size of 1000, and Tables 2-4 are with a buffer size of 5000. Could the authors either provide results for both buffer sizes consistently across all experiments or explain their rationale for using different buffer sizes in different analyses?
>
> We thank the reviewer for pointing out the inconsistency in buffer sizes across figures and tables. Initially, we used a buffer size of 1000 in Figures 3 and 5 to highlight the accuracy improvements achieved by SATCH under stricter memory conditions.
>
> To address this, we have updated Figures 3 and 5 to reflect results with a buffer size 5000, aligning them with the settings used in Tables 2–4. The original figures with a buffer size 1000 have been moved to the appendix. Due to time limitations, we will expand on running the ablation studies in Tables 2-4 in the future.
>
> > The choice of backbone: The backbones the authors tested are ResNet-18 and 3-layer convnet. Are there any potential challenges or modifications needed to apply SATCH to transformer-based architectures? Could the authors provide preliminary results with a transformer-based architecture like ViT if feasible?
>
> While we have yet to experiment with transformer-based backbones, SATCH may be effectively adapted to such architectures. We outline the considerations and how they can be integrated below:
>
> Transformer-based continual learning methods, such as Adapters [P3, P4], use task-specific bottleneck modules inserted at each transformer layer. These methods typically train an adapter per task and attempt to fuse them to promote knowledge transfer through replay and knowledge distillation methods. SATCH can be used to provide a different view during the fusion process to retain past knowledge more effectively.
>
> A key consideration is selecting an appropriate backbone for the assistant teacher. As shown in Table 3, the capacity gap [P7] between the assistant teacher and the student impacts the effectiveness of SATCH’s components. A lightweight vision transformer variant may be appropriate for transformer-based architectures as the assistant teacher backbone.
>
> We plan to explore SATCH’s integration into transformer-based frameworks such as ADA [P4]. Specifically, SATCH’s diverse logits may be incorporated into the fusion step during the distillation process, improving the diversity of knowledge transferred to the new adapter.

---

> ### Author Response · Authors · 2024-11-22
> **Response to Reviewer 7tBY (3/3)**
>
> > Also, it would be better to compare with more recent SOTA and other class incremental learning methods such as [1-3].
> [1] Class-Incremental Learning With Strong Pre-Trained Models, CVPR 2022 [2] DyTox: Transformers for Continual Learning With DYnamic TOken eXpansion, CVPR 2022 [3] Class-incremental learning with clip: Adaptive representation adjustment and parameter fusion, ECCV 2024
>
> Thank you for suggesting a comparison with more SOTA methods, such as [1-3]. Many SOTA methods use dynamic architectures or pre-trained models to achieve high accuracy. Below, we discuss the applicability of these methods to our setting.
>
> Dynamic architecture methods often come at the cost of increased model complexity, which can grow linearly with the number of tasks. For instance, on CIFAR100, a dynamic architecture method DER [P5] achieves 75.4% accuracy after learning all tasks, but the model parameters increased from 11M to 111M after ten tasks. In contrast, SATCH combined with DER++ achieves 59.97% accuracy using only 12M parameters after learning all tasks. Despite the accuracy improvements from DER and other dynamic architecture methods, it may not apply to real-world settings with limited memory.
>
> Pre-trained models may assume that the labels learned are correct. However, this may not be the case in the noisy class incremental setting [P8]. When using a method proposed by Wu et al. [1], which freezes network layers to preserve learned features, the model may be unable to recover from the noisy representations learned.
>
> Dynamic architecture techniques often rely on fusion [3] to reduce the model parameters. In future work, we will investigate whether SATCH reduces knowledge loss during fusion by integrating diverse task-specific knowledge with pre-trained and dynamic architecture methods.
>
> [P1] You, S., Xu, C., Xu, C., & Tao, D. (2017, August). Learning from multiple teacher networks. In Proceedings of the 23rd ACM SIGKDD international conference on knowledge discovery and data mining (pp. 1285-1294).
>
> [P2] Kim, S., Noci, L., Orvieto, A., & Hofmann, T. (2023). Achieving a better stability-plasticity trade-off via auxiliary networks in continual learning. In Proceedings of the IEEE/CVF Conference on Computer Vision and Pattern Recognition (pp. 11930-11939).
>
> [P3] Ermis, B., Zappella, G., Wistuba, M., Rawal, A., & Archambeau, C. (2022). Continual learning with transformers for image classification. In *Proceedings of the IEEE/CVF Conference on Computer Vision and Pattern Recognition* (pp. 3774-3781).
>
> [P4] Kim, G., Liu, B., & Ke, Z. (2022, November). A multi-head model for continual learning via out-of-distribution replay. In Conference on Lifelong Learning Agents (pp. 548-563). PMLR.
>
> [P5] Yan, S., Xie, J., & He, X. (2021). Der: Dynamically expandable representation for class incremental learning. In *Proceedings of the IEEE/CVF conference on computer vision and pattern recognition* (pp. 3014-3023).
>
> [P6] Wang, L., & Yoon, K. J. (2021). Knowledge distillation and student-teacher learning for visual intelligence: A review and new outlooks. IEEE transactions on pattern analysis and machine intelligence, 44(6), 3048-3068.
>
> [P7] Son, W., Na, J., Choi, J., & Hwang, W. (2021). Densely guided knowledge distillation using multiple teacher assistants. In Proceedings of the IEEE/CVF International Conference on Computer Vision (pp. 9395-9404).
>
> [P8] Sarfraz, F., Arani, E., & Zonooz, B. Error Sensitivity Modulation based Experience Replay: Mitigating Abrupt Representation Drift in Continual Learning. In The Eleventh International Conference on Learning Representations.

---

> ### Author Response · Authors · 2024-11-26
> **Gentle Reminder**
>
> We sincerely thank you once again for your valuable and constructive feedback. As a gentle reminder, the discussion period will close in approximately one week. We would be happy to further discuss any unresolved questions that you may have.

---

> > ### Comment · Reviewer_7tBY · 2024-11-27
> >
> > Thank you for your response. I appreciate the additional experiments and detailed explanation.
> >
> > However, some of my concerns are not fully addressed.
> >
> > > Diverse knowledge in our study refers to broader and more generalized feature representations
> >
> > Diverse knowledge distillation sounds like it can be useful for catastrophic forgetting but it's a global representation. These two are different terms.  The term, diverse knowledge is vague and can be misleading.
> >
> > The authors showed an example of the polar bear.
> >
> > > the assistant teacher highlights task-specific features, such as the ears and legs, while the main teacher focuses on more generalized features, such as the eyes.
> >
> > I'm not sure the eyes are more generalized parts than the ears and legs.
> >
> > After reading the rebuttal answers, the diverse knowledge distillation part is unclear, especially how this can help continual learning and reduce forgetting.
> >
> > Also, using a transformer-based architecture as a backbone can be a valuable experiment, and based on the authors' explanation, it seems feasible. However, the authors were not able to provide the preliminary results.
> >
> > Given the improvement, I will raise my score but cannot raise it further because of these reasons.

---

> > > ### Author Response · Authors · 2024-11-28
> > > **Thank you for your detailed feedback.**
> > >
> > > Thank you for your detailed feedback and for raising your score. We appreciate your insights and the opportunity to improve our work.
> > >
> > > > Diverse knowledge distillation sounds like it can be useful for catastrophic forgetting but it's a global representation. These two are different terms. The term, diverse knowledge is vague and can be misleading.
> > >
> > > To clarify, the term “diverse knowledge” is often used in multi-teacher distillation, following the principle of “two heads are better than one” [P3]. For instance, approaches such as averaging multiple teacher logits [P1] or randomly selecting a teacher for distillation [P2] improved student accuracy. Liu et al. [P3] further demonstrate that using teachers with varied backbones and training distributions improves accuracy by providing more informative guidance.
> > >
> > > We acknowledge that "diverse knowledge" may be interpreted in different ways. To improve clarity, we have updated the term to “complementary knowledge,” where the SATCH's assistant teacher is specifically designed to provide complementary knowledge to the main teacher, given the constraints of continual learning.
> > >
> > > In knowledge distillation, the main teacher is prone to forgetting task-specific information as it generalizes to wider variety of tasks. SATCH reduces the loss of task-specific knowledge during distillation by providing complementary task-specific knowledge. Inspired by Liu et al. [P3] we adapt our assistant teacher to the constraints of continual learning, SATCH’s assistant teacher retains task-specific knowledge by exclusively learning a single task and uses a smaller backbone to provide a different understanding of the task to reduce the forgetting of past knowledge.
> > >
> > > > I'm not sure the eyes are more generalized parts than the ears and legs.
> > >
> > > We appreciate your observation regarding the polar bear example. We agree that emphasizing features in isolation could be misleading. We intended to illustrate how the assistant teacher identifies features distinct from those of the main teacher. When their outputs are combined, the resulting feature map is broader and more complementary. We have revised the introduction (highlighted in blue) to avoid confusion.
> > >
> > > > Also, using a transformer-based architecture as a backbone can be a valuable experiment
> > >
> > > We agree that testing a transformer-based backbone is an important direction and are actively working on these experiments. Due to time constraints, we clarify the concept of “diverse knowledge” and hope to provide initial results before the discussion period ends.
> > >
> > > [P1] You, S., Xu, C., Xu, C., & Tao, D. (2017, August). Learning from multiple teacher networks. In Proceedings of the 23rd ACM SIGKDD international conference on knowledge discovery and data mining (pp. 1285-1294).
> > >
> > > [P2] Fukuda, T., Suzuki, M., Kurata, G., Thomas, S., Cui, J., & Ramabhadran, B. (2017, August). Efficient Knowledge Distillation from an Ensemble of Teachers. In Interspeech (pp. 3697-3701).
> > >
> > > [P3] Liu, Z., Liu, Q., Li, Y., Liu, L., Shrivastava, A., Bi, S., ... & Zhao, Z. (2024). Wisdom of Committee: Distilling from Foundation Model to SpecializedApplication Model. *arXiv preprint arXiv:2402.14035*.

---

> ### Author Response · Authors · 2024-12-03
> **SATCH Transformer Backbone Results**
>
> Thank you for waiting for the preliminary results, we apply SATCH to Mixture-of-Experts Adapters (MoE) [P1], which use LoRA to learn tasks incrementally while reducing runtime and memory overhead. While MoE reduces computational complexity compared to parameter-isolation methods, it suffers from forgetting due to an expert having to learn multiple tasks incrementally.
>
> By applying SATCH, we reduce forgetting by distilling past task knowledge into the model as new tasks are learned. We evaluate MoE and MoE + SATCH on CIFAR100 using a buffer size of 5000 and λ=0.1. The final accuracy is reported below:
>
> | Methods | Final Accuracy (%) |
> | --- | --- |
> | **MoE + SATCH (Ours)** | **79.78** |
> | MoE (Yu et al. CVPR 24) [P1] | 78.42 |
>
> SATCH improves MoE accuracy by 1.36%. This shows that SATCH can effectively improve accuracy of MoE that uses a pre-train transformer based architecture.
>
> To investigate forgetting in MoE adapters, we track accuracy changes over time for the first and second tasks:
>
> | Tasks Learned | 0 | 1 | 2 | 3 | 4 | 5 | 6 | 7 | 8 | 9 |
> | --- | --- | --- | --- | --- | --- | --- | --- | --- | --- | --- |
> | MoE Task 1 Accuracy | **98.5** | 95.3 | 91.0 | 85.5 | 83.0 | 82.3 | 81.3 | 77.4 | 80.1 | 80.5 |
> | + SATCH Task 1 Accuracy | 98.3 | **97.3** | **95.5** | **91.4** | **88.8** | **87.7** | **88.1** | **87.5** | **85.1** | **84.7** |
> | MoE Task 2 Accuracy | **93.2** | 90.3 | 88.1 | 87.4 | 85.5 | 82.5 | 78.9 | 79.6 | 76.9 |  |
> | + SATCH Task 2 Accuracy | 92.6 | **92.6** | **91.0** | **90.5** | **86.0** | **85.9** | **84.2** | **83.7** | **81.0** |  |
>
> These results demonstrate that SATCH reduces forgetting compared to MoE alone, retaining higher accuracy for earlier tasks as more tasks are learned. The backbone used by MoE is `ViT-B/32` , to implement SATCH we use the same backbone for SATCH’s assistant teacher. For future work, we investigate different backbones such as `ViT-B/16` and aim to apply SATCH to further transformer based methods
>
> [P1] Yu, J., Zhuge, Y., Zhang, L., Hu, P., Wang, D., Lu, H., & He, Y. (2024). Boosting continual learning of vision-language models via mixture-of-experts adapters. In *Proceedings of the IEEE/CVF Conference on Computer Vision and Pattern Recognition* (pp. 23219-23230).

---

> > ### Comment · Reviewer_7tBY · 2024-12-03
> >
> > Thank you for providing the transformer-based results. The earlier task accuracies are improved by more than 4%, but the final accuracy improvement is only 1.36%. This means that the later task(s) are underperforming. I'm unsure if this comes from the model change or the method itself, but this is a big weakness for practical applications. I'd like to hear from the authors about it if possible.

---

> ### Author Response · Authors · 2024-12-03
> **Deeper dive into all task accuracies over time**
>
> Thank you, we dig deeper and discuss this observation. We show the results for all task progressions below, in the paper we will use a graph for easier interpretability:
> | Tasks Learned | 0 | 1 | 2 | 3 | 4 | 5 | 6 | 7 | 8 | 9 |
> | --- | --- | --- | --- | --- | --- | --- | --- | --- | --- | --- |
> | MoE Task 1 Accuracy | **98.5** | 95.3 | 91.0 | 85.5 | 83.0 | 82.3 | 81.3 | 77.4 | 80.1 | 80.5 |
> | + SATCH Task 1 Accuracy | 98.3 | **97.3** | **95.5** | **91.4** | **88.8** | **87.7** | **88.1** | **87.5** | **85.1** | **84.7** |
> | MoE Task 2 Accuracy | **93.2** | 90.3 | 88.1 | 87.4 | 85.5 | 82.5 | 78.9 | 79.6 | 76.9 |  |
> | + SATCH Task 2 Accuracy | 92.6 | **92.6** | **91.0** | **90.5** | **86.0** | **85.9** | **84.2** | **83.7** | **81.0** |  |
> | MoE Task 2 Accuracy | **87.3** | 82.4 | 77.6 | 75.2 | 69.1 | 72.8 | 65.7 | 61.6 |
> | + SATCH Task 2 | 83.2 | **82.8** | **79.2** | **80.2** | **77.0** | **76.0** | **73.7** | **63.4** |
> | MoE Task 3 Accuracy | **85.5** | 82.1 | 81.4 | 78.0 | 75.3 | 74.2 | 76.0 |
> | + SATCH Task 3 | 81.5 | **85.2** | **83.1** | **79.9** | **80.8** | **80.2** | **78.2** |
> | MoE Task 4 Accuracy | **90.8** | **88.3** | **90.7** | 83.3 | 81.8 | 79.4 |
> | + SATCH Task 4 | 87.9 | 88.0 | 89.2 | **86.7** | **84.5** | **81.0** |
> | MoE Task 5 Accuracy | **91.5** | **90.5** | 86.9 | 88.5 | 88.5 |
> | + SATCH Task 5 | 87.1 | 88.5 | **89.4** | **88.7** | **88.6** |
> | MoE Task 6 Accuracy | **78.0** | **77.8** | **74.8** | 72.6 |
> | + SATCH Task 6 | 72.5 | 74.1 | 73.4 | **74.8** |
> | MoE Task 7 Accuracy | **88.9** | **84.7** | 83.9 |
> | + SATCH Task 7 | 81.8 | 82.8 | 83.9 |
> | MoE Task 8 Accuracy | **87.2** | **86.1** |
> | + SATCH Task 8 | 84.7 | 84.8 |
> | MoE Task 9 Accuracy | **78.7** |
> | + SATCH Task 9 | 76.4 |
>
> From the table above, we observe that SATCH provides more stable accuracy across tasks, with less variation as new tasks are learned compared to MoE. This stability is due to knowledge distillation, which applies functional regularization to reduce forgetting. In contrast, MoE achieves higher initial accuracy on later tasks, but its accuracy declines over time as it optimizes weights exclusively for the current task at the expense of previous tasks. This effect of higher initial accuracy but lower final accuracy is also observed on the ResNet-18 backbone as seen in Figure 3.
>
> Notably, SATCH exhibits slightly lower accuracy on the final two tasks, as fewer or no additional tasks have been learned to cause forgetting. This trade-off, between remembering old tasks (`stability`) and learning new tasks (`plasticity`), is a common challenge in continual learning [P1]. However, given the trends of the earlier tasks, MoE only has higher accuracy initially and shows lower accuracy retention over time as more tasks are learned.
>
> For practical applications requiring higher plasticity, the weight of the knowledge distillation loss can be adjusted to increase current-task learning while preserving reasonable stability. This enables SATCH to offer a flexible balance between stability and plasticity depending on the application’s needs.
>
> [P1] Mermillod, M., Bugaiska, A., & Bonin, P. (2013). The stability-plasticity dilemma: Investigating the continuum from catastrophic forgetting to age-limited learning effects. *Frontiers in psychology*, *4*, 504.

---

### Meta-Review · Area_Chair_USqf · 2024-12-23

**Metareview:**

The paper presents a novel class-incremental continual learning framework designed to mitigate catastrophic forgetting through a specialized assistant-teacher mechanism. The assistant teacher is trained on individual tasks prior to the student, providing task-specific guidance that enhances memory retention and reduces forgetting of previously learned tasks. Experimental results on benchmarks such as CIFAR-100, TinyImageNet, and MiniImageNet show improvements in continual learning accuracy.

The paper's strengths include its compatibility with existing class-incremental learning methods and its clear, easy-to-follow writing. However, the work has notable weaknesses, including an insufficient literature review and a lack of robust comparisons with key baselines.

Given these limitations, I recommend rejecting this submission.

**Additional Comments On Reviewer Discussion:**

During the discussion period, all reviewers actively engaged with the authors; however, none were fully satisfied with the responses provided.

After carefully reviewing the concerns raised by the reviewers, I found that the authors failed to adequately address two critical issues: (1) unfair comparisons, as highlighted by Reviewer yDJ1, and (2) a lack of empirical comparisons with key baselines, a concern raised by all reviewers.

These two issues are closely intertwined, as the proposed method relies on rehearsal data but achieves performance that falls short of existing rehearsal-free methods combined with pre-trained models. Unfortunately, the authors did not offer a substantial response to this concern and failed to include an in-depth literature review or performance comparison with state-of-the-art methods, such as HidePrompt, CPrompt, and others.

These unresolved issues significantly undermine the validity and potential impact of the paper, limiting its relevance to the broader research community. As such, I believe the submission does not meet the high standards required for acceptance at the prestigious ICLR conference.

---

### Decision · Program_Chairs · 2025-01-22

Reject